

# Emergent orbital magnetization in Kitaev quantum magnets

**Saikat Banerjee[1] and Shi-Zeng Lin[1,2,3]**

**1** Theoretical Division, T-4, Los Alamos National Laboratory,
Los Alamos, New Mexico 87545, USA
**2** Theoretical Division, CNLS, Los Alamos National Laboratory,
Los Alamos, New Mexico 87545, USA
**3** Center for Integrated Nanotechnologies (CINT), Los Alamos National Laboratory,
Los Alamos, New Mexico 87545, USA

## Abstract

**Unambiguous identification of the Kitaev quantum spin liquid (QSL) in materials remains a huge challenge despite many encouraging signs from various measurements. To facilitate the experimental detection of the Kiteav QSL, here we propose to use remnant charge response in Mott insulators hosting QSL to identify the key signatures of QSL. We predict an emergent orbital magnetization in a Kitaev system in an external magnetic field. The direction of the orbital magnetization can be flipped by rotating the external magnetic field in the honeycomb plane. The orbital magnetization is demonstrated explicitly through a detailed microscopic analysis of the multiorbital Hubbard-Kanamori Hamiltonian and also supported by a phenomenological picture. We first derive the localized electrical loop current operator in terms of the spin degrees of freedom. Thereafter, utilizing the Majorana representation, we estimate the loop currents in the ground state of the chiral Kitaev QSL state, and obtain the consequent current textures, which are responsible for the emergent orbital magnetization. Finally, we discuss the possible experimental techniques to visualize the orbital magnetization which can be considered as the signatures of the underlying excitations.**



# 1  Introduction

Understanding quantum spin liquid (QSL) states and identifying them [1–4] in materials, has become a centerpiece of research in modern condensed matter physics. The QSL is a topologically ordered state with long-range entanglement that appears in spin systems with frustrated magnetic interactions. Although the beginning of the exploration of various QSL states dates back to the seminal work by Anderson on resonating valence bond states in antiferromagnetic Mott insulators [5], the research in QSLs rapidly intensified with the discovery of high-$T_c$ cuprates [6] due to its possible connection to the origin of superconductivity [7]. Over the last four decades, various theoretical proposals [8–11] have been made to understand this exotic phase of matter. All these works hinge on a universal consensus about the fractional nature of quasiparticles in a typical QSL state. However, an unambiguous material-based realization of a QSL phase is hitherto absent.

    Among the zoo of various predicted QSL states, the Kitaev spin liquid [12] plays a special role. It is an exactly solvable $S = 1/2$ spin model on a two-dimensional honeycomb lattice, in which the spins fractionalize into (i) Majorana fermions and (ii) $\mathbb{Z}_2$ vortices, also known as visons. Recently, substantial interest has emerged in the study of the Kitaev model owing to its potential realization in a few spin-orbit coupled Mott insulators such as complex iridates $A_2IrO_3$ [13] (A = Na, Li) and $\alpha$-RuCl$_3$ [14]. For real materials, however, non-Kitaev interactions and further spin-exchange interactions are inevitably present [15–18], which spoils the integrability of the ideal Kitaev model and makes it difficult to assess its predictions. In

this regard, various proposals have been put forward to tune the magnetic exchange interactions towards the ideal Kitaev limit by applying strain [19], modifying spin-orbit coupling (SOC) [20] or through Floquet engineering [21–24] in the honeycomb Mott insulators.

Recent experiments on $\alpha$-RuCl$_3$ [25–34] have provided a strong evidence for the existence of the Kitaev QSL phase in an intermediate magnetic field regime of $B \sim 8 - 14$ T. Namely, nuclear magnetic resonance [25] and neutron diffraction [26] experiments observed field-induced spin gap opening around 10 T consistent with the physics of an ideal Kitaev model in an applied magnetic field. Particular features of such gap opening have also been speculated by inelastic neutron scattering studies [29], although the similar evidence of gap opening can be associated with a partially polarized magnetic phase at high magnetic field. Specifically, the observation of half-quantized thermal Hall conductivity [34, 35] has been interpreted as a signature of chiral Majorana edge modes in $\alpha$-RuCl$_3$. On the other hand, the observation of thermal conductivity oscillations in $\alpha$-RuCl$_3$ as a function magnetic field, for an intermediate magnetic-field regime, suggests a metallic nature of the underlying quasiparticles, despite that the material is a perfect Mott insulator with a large Mott gap [32]. Nonetheless, direct experimental observation of the Kitaev phase has remained somewhat controversial [36]. The reason is that these experimental probes provide, mostly, an indirect signature of the underlying excitations of the Kitaev model, and cannot unambiguously determine the nature of the quasiparticles. Hence, the current *state-of-the-art* research demands an alternative, more direct, way to detect the Majorana and vison excitations in the Kitaev QSL phase.

Motivated by these fascinating experimental findings, here, we consider a different route to *electrically* detect the signatures of the fractionalized *neutral* excitations in the Kitaev model. A priori, such an approach might seem counter-intuitive as the parent compounds, realizing proximate Kitaev physics, are Mott insulators. However, in the paper by Bulaevskii *et al.*, it was shown that certain frustrated Mott insulators can exhibit non-zero *spontaneous circular electrical currents* or *nonuniform charge distribution* in the ground state [37]. On the other hand, spontaneous breakdown of injected electrons on a QSL material, into the fractionalized excitations, can be used to detect Majorana fermions [38]. A previous theoretical work [39] has recently proposed an experimental set-up utilizing the local charge distribution to detect vison excitations in $\alpha$-RuCl$_3$; whereas another contemporary work [40] proposed a more exotic electrical access to the Majorana fermions in Kitaev materials utilizing the transmutation protocol [38, 41, 42].

In this continuing search for various innovative electrical access in Mott insulators, one very exciting aspect still remains unexplored – *the induced electrical loop currents*, which is allowed in QSL without time-reversal symmetry. The orbital coupling between the emergent gauge field and physical gauge field for QSL with spinon Fermi surface was discussed before [43]. Here, we develop a theoretical formalism to analyze such localized orbital loop current profile in a multiorbital Mott insulator relevant for $\alpha$-RuCl$_3$ and propose experimental set-ups to detect the signatures of Majorana and vison excitations in a Kitaev quantum magnet. For a half-filled single orbital Hubbard model on a 2D lattice with $|t_{ij}| \ll U$, with an on-site Coulomb repulsion $U$ and the inter-site hopping amplitude $t_{ij}$, the corresponding expression for the loop current operator reads as [37]

$$\mathcal{I}_{ij,k} = \frac{24 t_{ij} t_{jk} t_{ki}}{U^2} \mathbf{S}_k \cdot (\mathbf{S}_i \times \mathbf{S}_j), \tag{1}$$

where $\mathbf{S}_i = (S_i^x, S_i^y, S_i^z)$ is the spin-1/2 operator, and $(ijk)$ labels the site indices on the smallest triangular plaquette within the lattice. Eq. (1) can be understood from symmetry consideration. Both the spin chirality $\mathbf{S}_k \cdot (\mathbf{S}_i \times \mathbf{S}_j)$ and current are odd under time reversal operation and spatial inversion (i.e. interchange of the indices $i$ and $j$). The system also has SU(2) spin rotation symmetry. Unless explicitly mentioned, we use natural units $\hbar = e = c = 1$ throughout this paper.

In this work, we investigate the induced localized loop current profile in a multiorbital spin-orbit coupled Hubbard-Kanamori model relevant for Kiteav quantum magnets such as $\alpha$-RuCl$_3$. The key findings of our work are listed as follows: (i) Utilizing the Schrieffer-Wolff transformation (SWT), we derive the expression for the localized loop current operator in the Mott insulator to the third order in perturbation expansion [see Eq. (6a)]. The breakdown of the SU(2) spin rotation symmetry, due to spin-orbit coupling, leads to a different loop current expression compared to its single-orbital counterpart.

(ii) In the presence of an external magnetic field, the Kitaev ground state hosts non-zero expectation value of such localized loop currents. Adopting microscopic parameters relevant for $\alpha$-RuCl$_3$ [44,45], we obtain the orbital current profile in a finite 2D honeycomb lattice with localized edge and bulk loop current distributions, as shown in Fig. 2. The induced localized currents in the bulk of a 2D honeycomb lattice constitute an emergent super-structure of two inter-penetrating triangular lattices as illustrated in Fig. 2(b,b′). When an in-plane [$ab$-plane in Fig. 1(a)] magnetic field is applied to the Kitaev system, these emergent orbital currents produce an out-of-plane ($c$-axis) magnetic field, which we identify as the *key feature* of a Kitaev system.

(iii) Finally, we notice that vison excitations drastically modify the current distribution profile and the associated orbital magnetization locally, as shown in Fig. 3. We provide quantitative estimates for this change and about the possible measurement of such orbital magnetization using various possible experimental techniques. Our work provides an important platform to directly detect the electrical signatures of the concomitant excitations (Majorana and vison) in a Kitaev quantum magnet.

## 2 Phenomenological picture

Before moving to full microscopic model analysis, here we provide a more intuitive picture of why we expect an orbital magnetization in Mott insulators hosting the Kitaev quantum spin liquid in terms of the parton theory [46,47]. The electron operator can be written as $c_\sigma = bf_\sigma$, where $b$ is a boson operator carrying electron charge quantum number and $f_\sigma$ is the fermionic spinon operator carrying spin quantum number. The parton decomposition leads to an emergent gauge field $\mathbf{a}$, which inherently couple to the spinons as $f_\sigma \rightarrow f_\sigma \exp(ia)$, while the charged boson couple to both the physical gauge field $\mathbf{A}$ and emergent gauge field $\mathbf{a}$, as $b \rightarrow b \exp[i(A-a)]$. The effective low-energy Lagrangian for the $b$ boson has the standard Ginzburg-Landau form [48]

$$\mathcal{L}_b = \sum_{\mu=x,y,z} |(i\partial_\mu + a_\mu - A_\mu)b|^2 - g|b|^2 - \frac{u}{2}|b|^4 + \cdots \tag{2}$$

Here we only keep the spatial components of the gauge fields, because they are responsible for the diamagnetic response discussed below. When $b$ boson condenses for $g < 0$, the Anderson-Higgs mechanism generates a term proportional to $(a_\mu - A_\mu)^2$, which locks $\mathbf{a}$ to $\mathbf{A}$ [10]. As a consequence, the emergent gauge field behaves the same way as the physical gauge field, and the system is metal if $f_\sigma$ forms the fermi surface or a superconductor if $f_\sigma$ forms Cooper pairs and condenses.

In the Mott insulator, $b$ boson is gapped with $g > 0$. However, there is still diamagnetic response in $\mathbf{A} - \mathbf{a}$ due to the gapped charged boson, similar to the Landau diamagnetism in metal, albeit weaker. When an external magnetic is applied in Mott insulators, local current loops are induced which generate magnetization opposite to the applied field as a diamagnetic response. The diamagnetic response increases with decreasing charge gap, and hence the effect discussed here is more prominent near the Mott transition. The effective Lagrangian of

the system after integrating out $b$ boson has the form [49]

$$\mathcal{L} = \mathcal{L}_f(f_\sigma, \mathbf{a}) - \frac{\chi_b}{2}[\nabla \times (\mathbf{a} - \mathbf{A})]^2 - \frac{\chi_B}{2}(\nabla \times \mathbf{A})^2, \tag{3}$$

where $\chi_b$ accounts for the diamagnetic susceptibility due to the gapped boson $b$, $\chi_B$ is the susceptibility of the background. $\mathcal{L}_f(f_\sigma, \mathbf{a})$ is the Lagrangian describing the spinons coupled to the emergent gauge field $\mathbf{a}$. Due to the diamagnetic response, an emergent magnetic field induces a physical magnetic field, which is obtained by minimizing $\mathcal{L}$ with respect to $\mathbf{B} \equiv \nabla \times \mathbf{A}$: $\mathbf{B} = (\nabla \times \mathbf{a})\chi_b/(\chi_b + \chi_B)$. We remark one subtle point that $\mathbf{a}$ is compact meaning that the system is invariant under $\mathbf{a} \rightarrow \mathbf{a} + 2\pi$. Because of the compactness of $\mathbf{a}$, the flux of $\mathbf{a}$ is defined upto modulo of $2\pi$. In the context Kitaev Mott insulator, therefore a vison carrying $\pi$ flux without magnetic field is time-reversal symmetric and does not induce a nonzero $\mathbf{B}$. Here we have considered isotropic response. For the two-dimensional systems we consider below, the diamagnetic susceptibility is only for the field component perpendicular to the plane, and the induced $\mathbf{B}$ is also along the direction perpendicular to the plane.

In the spinon description, the chiral Kitaev quantum spin liquid stabilized by a magnetic field corresponds to the state with $f_\sigma$ fermions being in the $p + ip$ superconducting state, where the time-reversal symmetry is explicitly broken by the magnetic field [50]. In this phase, there exists chiral edge current of $f_\sigma$ and the associated magnetization of $\nabla \times \mathbf{a}$. At the same time, the vortex carries flux of $\mathbf{a}$. Because of the diamagnetism due to the gapped $b$ boson, the emergent magnetic field induces a physical magnetic field. This picture will be corroborated through the explicit calculation of a microscopic model below.

# 3 Microscopic model

We start from the microscopic multiorbital description for $\alpha$-RuCl$_3$, however, this formalism can be suitably generalized to other Kitaev candidate materials. The electronic configuration of the transition metal (TM) ion Ru in $\alpha$-RuCl$_3$ is $4d^5$. These *five* electrons reside in the $t_{2g}$ manifold formed by the three orbitals: $d_{xy}$, $d_{yz}$ and $d_{zx}$. The presence of strong atomic SOC [51–56] further splits this manifold into $J_{\text{eff}} = 1/2$ and $J_{\text{eff}} = 3/2$ states. This leads to a completely filled $J_{\text{eff}} = 3/2$ manifold with the four electrons, and the remaining electron resides in the $J_{\text{eff}} = 1/2$ sector. Consequently, it leads to an effective one-hole model, constructed with the following parameters: an on-site Coulomb repulsion of strength $U$, the Hund's coupling $J_{\text{H}}$, and (next-)nearest neighbor hopping $(t_2')$ $t_2$, respectively [18,52] [see Fig. 1(a,b)].

In the strong-coupling limit ($t_2, t_2' \ll U, J_{\text{H}}$), the low-energy effective Hamiltonian is obtained by the (SWT) [see Appendix C] and leads to the ferromagnetic Kitaev Hamiltonian as

$$\mathcal{H}_0 = -K \sum_{\langle ij \rangle; \alpha} S_i^\alpha S_j^\alpha, \quad K = \frac{8J_{\text{H}} t_2^2}{3(U^2 - 4UJ_{\text{H}} + 3J_{\text{H}}^2)}, \tag{4}$$

where $\langle ij \rangle$ denotes the nearest-neighbor sites, and $\alpha \in \{x, y, z\}$ labels the planar orientation of the bond-type as illustrated in Fig. 1(a,b). The Kitaev coupling $K$ in Eq. (4) is obtained to the lowest order in perturbation, and does not include $t_2'$. Note that we treat $t_2'$ as a small perturbation. We first obtain the ground state configuration without $t_2'$. Then the induced orbital current is obtained by considering $t_2'$ as a small perturbation, which is accurate up to the first order in $t_2'$.

This model can be solved exactly, as shown by Kitaev himself [12], by introducing Majorana representation of the spin-1/2 operators as $S_i^\alpha = i b_i^\alpha c_i/2$ for $\alpha = x, y, z$. Here $b_i^\alpha, c_i$ are four mutually anti-commuting Majorana operators. In terms of $b_i^\alpha$ and $c_i$, the Hamiltonian in Eq. (4) can be rewritten as $\mathcal{H}_0 = iK/4 \sum_{\langle ij \rangle} u_{ij} c_i c_j$, where $u_{ij}(= i b_i^\alpha b_j^\alpha)$ are the $\mathbb{Z}_2$ gauge

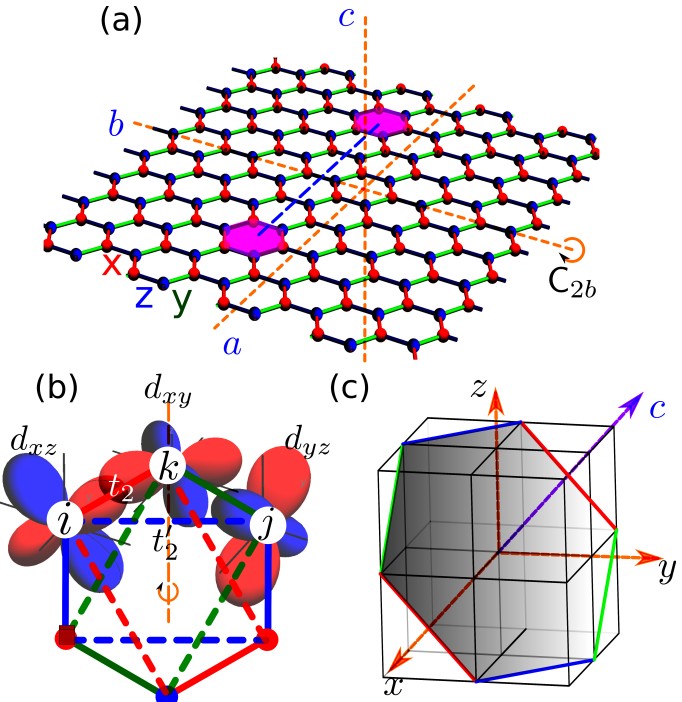

Figure 1: (a) Schematic of a finite size ($L \times L$) honeycomb lattice made out of magnetic ions with two $\mathbb{Z}_2$ vortices (vison excitations) [*shaded region*]. The vison excitations are separated by a string operator (*dashed blue line*) which flips $u_{ij}$ eigenvalue when it crosses a particular bond $\langle ij \rangle$. The colored bonds represent three anisotropic bond-dependent Kitaev interactions. The $\pi$-rotation symmetry along the crystalline axes $b$ is illustrated by the dashed orange lines. There is an additional three-fold rotation symmetry along the crystalline $c$-axis. Note that the $\pi$-rotation along the $a$-axis is not a symmetry in the presence of ligands. As an example, the crystal symmetry for the monolayer $\alpha$-RuCl$_3$ is P6$_3$/mcm which does not contain $\pi$-rotation along the $a$-axis as a symmetry. (b) A single hexagonal plaquette with the (next)- nearest neighbor hopping amplitudes ($t_2'$) $t_2$ formed by the bond-dependent orbital overlaps. (c) The relative orientation of the honeycomb plane ($abc$-coordinate system) in panel (a) with respect to the spin quantization axes ($xyz$-coordinate system).

field operators with eigenvalues $\pm 1$. Since $u_{ij}$ commutes with the Hamiltonian, one can pick a particular choice of $u_{ij}$ (a gauge sector $|\mathcal{G}\rangle$) and solve the remaining non-interacting Majorana Hamiltonian. The pinned flux in each honeycomb plaquette is obtained by the gauge-invariant loop operator $\mathcal{W}_p = \prod_{\langle ik \rangle \in \bigcirc_p} u_{ik}$. Here $\mathcal{W}_p = +1$ corresponds to zero-flux in a plaquette, whereas $\mathcal{W}_p = -1$ signifies a $\pi$-flux and corresponds to a $\mathbb{Z}_2$ vortex or a vison excitation [see the shaded hexagons in Fig. 1(a)]. Following Lieb's theorem [57], the ground state of the Majorana fermions is obtained with the gauge choice of $\mathcal{W}_p = 1$, $\forall p$, for all the hexagonal plaquettes. Starting from a uniform gauge configuration, one may create a single vison excitation by changing the eigenvalue of a single plaquette loop operator to $-1$.

To induce orbital magnetization, it is necessary to break the related symmetries. In crystals realizing the Kitaev model, the symmetry is low because of the spin-orbit coupling. As shown in Fig. 1(a), the system has translation symmetry, two-fold rotation symmetry along crystalline $b$-axis denoted by $C_{2b}$ in panel (a), 3-fold rotation symmetry along the $c$ axis, $C_{3c}$, time-reversal symmetry (TRS) and inversion symmetry. The seemingly two-fold rotation along the $a$-axis Fig. 1(a) is absent when one embeds the honeycomb plane into the parent crystal. To allow for orbital magnetization along the $c$ axis, we need to break the TRS and also the $C_{2b}$

symmetry. This can be achieved by applying a magnetic field with non-zero component perpendicular to the $b$ axis. One particularly interesting case is when the magnetic field is applied in the $a$-$b$ plane, which induces orbital magnetization along the $c$ axis. Generally, the induced magnetization is much weaker than the applied magnetic field. The case with an external magnetic field in the plane may be more convenient for experimental detection of the induced orbital magnetization.

## 4 Circulating loop currents

Since the parent compound of a Kitaev spin liquid is a multiorbital spin-orbit coupled Mott insulator, there is remnant charge response and there can exist localized circulating electrical currents. The charge response is governed by the charge excitation gap, which is of the order of $U$. Similar to the derivation of the effective spin Hamiltonian in Eq. (4) from the Hubbard model, we perform a strong-coupling expansion (SWT) for the current operator defined on the bond $\langle ij\rangle$ [see Fig. 1(b)] as

$$\mathcal{I}_{ij} = \frac{iet_2'\hat{\mathbf{r}}_{ij}}{\hbar} \sum_{\alpha,\beta,\sigma} \left( d_{j\alpha\sigma}^\dagger d_{i\beta\sigma} - d_{i\beta\sigma}^\dagger d_{j\alpha\sigma} \right), \tag{5}$$

where $d_{i\alpha\sigma}^\dagger$ creates an electron in the $i$-th site with spin $\sigma$ and orbital $\alpha, \beta \in \{x, y, z\}$, and obtain a generalization of Eq. (1) for the triangular loop current as (see Appendix B for the details of the derivation)

$$\tilde{\mathcal{I}}_{ij,k} = \mathcal{I}_0(3S_i^x S_j^y + S_i^y S_j^x)S_k^z + \mathcal{I}_0(3S_i^y S_j^z - 5S_i^z S_j^y)S_k^x + \mathcal{I}_0(3S_i^z S_j^x - 5S_i^x S_j^z)S_k^y, \tag{6a}$$

$$\mathcal{I}_0 = \frac{e\hat{\mathbf{r}}_{ij}}{\hbar} \frac{8t_2^2 t_2' J_H(U - 2J_H)}{9(U^2 - 4UJ_H + 3J_H^2)^2}, \tag{6b}$$

where $S_i^\alpha$ is the $\alpha$-th component of the spin at site $i$, and $U, J_H$, and $t_2$ are the microscopic parameters as defined earlier. For the current operator defined on the other bonds, we need to modify the loop current expression by permuting both the orbital and site indices on the three spin operators. The expression (6a) is consistent with the $C_{2b}$ symmetry [orange dashed line in Fig. 1(b)]. Under $C_{2b}$, $i \leftrightarrow j$, $k \leftrightarrow k$, $S^x \to -S^y$, $S^y \to -S^x$, $S^z \to -S^z$ and $\tilde{\mathcal{I}}_{ij,k} = -\tilde{\mathcal{I}}_{ji,k}$. However, the $C_{2b}$ symmetry cannot uniquely determine the form of $\tilde{\mathcal{I}}_{ij,k}$. Actually the absence of any terms with repeated index in the spin space (i.e. $S_i^x S_j^x S_k^z$) in Eq. (6a) is an artifact of considering an ideal situation, where we retain only $t_2$, and $t_2'$ hoppings in the multiorbital tight-binding (TB) description. Indeed, a more realistic TB modeling [18] for Kitaev quantum magnets would lead to additional three-spin terms with possible repeated indices. However, for the subsequent analysis in this work, we focus on the ideal Kitaev limit.

For an estimate of the amplitude of the induced electrical current, we adopt all the parameters entering Eq. (6b) from the recent *ab initio* [44] and *photoemission* reports [45] for $\alpha$-RuCl$_3$ as $U = 3.0$ eV, $J_H = 0.45$ eV, $t_2 = 0.191$ eV, and $t_2' = -0.058$ eV. The value of $t_2'$ has been adopted from Ref. [58]. Plugging in the numbers in Eqs. (4) and (6b), we estimate $|\mathcal{I}_0| \sim 30$ nA, with a Kitaev coupling $K \sim 0.023$ eV.

The key quantity now is the expectation value of the induced loop current operator in the ground state of the Kitaev model. The pure Kitaev model can be solved exactly from the Hamiltonian $\mathcal{H}_0 = iK/4 \sum_{ij} c_i c_j$, with all $u_{ij}$ chosen to be $+1$. The ground state eigenfunction can be written as $|\Psi_0\rangle = \mathcal{Z}|\mathcal{G}_0\rangle \otimes |\mathcal{M}\rangle$ where $|\mathcal{G}_0\rangle$ corresponds to $\mathcal{W}_p = 1$ for all hexagonal plaquettes and $|\mathcal{M}\rangle$ is the ground state of *gapless* Majorana fermions (see Appendix D). However as the Hilbert space is enlarged due to the fractionalization of the spin degrees of freedom, we need to project the eigenstates of $\mathcal{H}_0$ to the physical Hilbert space of the spin Hamiltonian in

Eq. (4). The corresponding projection is achieved by the local constraint $\mathcal{D}_i = ic_i b_i^x b_i^y b_i^z = 1$ and the global projection operator $\mathcal{Z} = \Pi_i(1 + \mathcal{D}_i)/2$ [12,59,60].

Due to the static nature of the vison excitations in a pure Kitaev model, only the first term, $S_i^x S_j^y S_k^z$, in Eq. (6a) contributes to the computation of the loop current expectation value in the Kitaev ground state. All the other five terms in Eq. (6a) do not contribute as they have zero expectation value in $|\Psi_0\rangle$ (see Appendix E). It is straightforward to see that the loop current expectation value vanishes in the pure Kitaev model ground state, i.e., $\langle\Psi_0|\tilde{\mathcal{I}}_{ij,k}|\Psi_0\rangle = 0$. This absence is not a surprise, since any finite loop current would lead to an orbital magnetization that breaks TRS, whereas the ground state $|\Psi_0\rangle$ preserves the TRS.

## 4.1 External magnetic field

The TRS of the system is broken in the presence of an external magnetic field. In this case, we may expect a finite loop current expectation value in the ground state. We consider the Kitaev model in an external magnetic field and explore its consequences. The integrability of the model is destroyed in the presence of an external magnetic field $\mathbf{h} = (h_x, h_y, h_z)$, defined along spin quantization axes (we call it the $xyz$-coordinate system). For a small $|\mathbf{h}|$, we can treat the effect of $\mathbf{h}$ in a perturbative manner. The lowest order perturbation correction, that breaks the TRS [12], reads as $\mathcal{H}_{\text{eff}} = -\kappa \sum_{ijk,\triangle} S_i^x S_j^y S_k^z$, where $\kappa = h_x h_y h_z/K^2$. Note that $\mathcal{H}_{\text{eff}}$ has the same form as the first term in $\tilde{\mathcal{I}}_{ij,k}$, and therefore it is natural to have a nonzero current in the QSL. Choosing the same gauge configuration $|\mathcal{G}_0\rangle$ as before, the total Hamiltonian $\mathcal{H} = \mathcal{H}_0 + \mathcal{H}_{\text{eff}}$ can be simplified as

$$\mathcal{H} = \frac{iK}{4} \sum_{\langle ik\rangle} c_i c_k + \frac{i\kappa}{8} \sum_{\langle\langle ij\rangle\rangle} c_i c_j \,, \tag{7}$$

where all the $u_{ij}$'s are replaced by their eigenvalues. The ground state eigenfunction of the above Hamiltonian can be written as $|\Psi_{\text{mag}}\rangle = \mathcal{Z}|\mathcal{G}_0\rangle \otimes |\mathcal{M}_{\text{mag}}\rangle$, where $|\mathcal{M}_{\text{mag}}\rangle$ is the ground state of the *gapped* Majorana fermions (see Appendix D.1), and $\mathcal{Z}$ is the global projection operator, as defined earlier. The loop current expectation value becomes non-vanishing in this case and $\langle\Psi_{\text{mag}}|\tilde{\mathcal{I}}_{ij,k}|\Psi_{\text{mag}}\rangle \propto \kappa$ for $\kappa \ll K$ (see Appendix D).

One key feature of the Kitaev system is that even an in-plane magnetic field [applied in the $ab$-plane shown in Fig. 1(b)] can have non-zero components along all the three spin quantization axes. Let us write the external magnetic field $\mathbf{h}^{\text{ext}}$ in the coordinate system containing the honeycomb plane (we call it the $abc$-coordinate system) as $\mathbf{h}^{\text{ext}} = h(\sin\theta\cos\phi, \sin\theta\sin\phi, \cos\theta)$, where $\phi$ and $\theta$ are azimuthal angle measured from the crystalline $a$ axis and polar angle from the crystalline $c$ axis, respectively, and h is the strength of the applied magnetic field. After a straightforward transformation between the $abc$- and the $xyz$-coordinate system [see Fig. 1(c)], we obtain (see Appendix C)

$$(h_x, h_y, h_z) = h\left(\frac{\cos\theta}{\sqrt{3}} + \frac{\sin\theta\cos\phi}{\sqrt{6}} - \frac{\sin\theta\sin\phi}{\sqrt{2}},\right.$$
$$\left.\frac{\cos\theta}{\sqrt{3}} + \frac{\sin\theta\cos\phi}{\sqrt{6}} + \frac{\sin\theta\sin\phi}{\sqrt{2}}, \frac{\cos\theta}{\sqrt{3}} - \sqrt{\frac{2}{3}}\sin\theta\cos\phi\right). \tag{8}$$

It is evident from Eq. (8) that even for an in-plane magnetic field (with $\theta = 0$), one can have non-zero $(h_x, h_y, h_z)$, leading to a finite $\kappa$. By rotating the magnetic field in the $ab$-plane, $\kappa$ can change the sign and even become zero for certain angles (see Appendix C for the details about the variation of $\kappa$ as a function of this rotation angle). For the field applied along the $b$ axis, $\theta = \pi/2$ and $\phi = \pi/2$, we have $\kappa = 0$ and therefore the induced orbital magnetization vanishes, which is consistent with the symmetry analysis above.

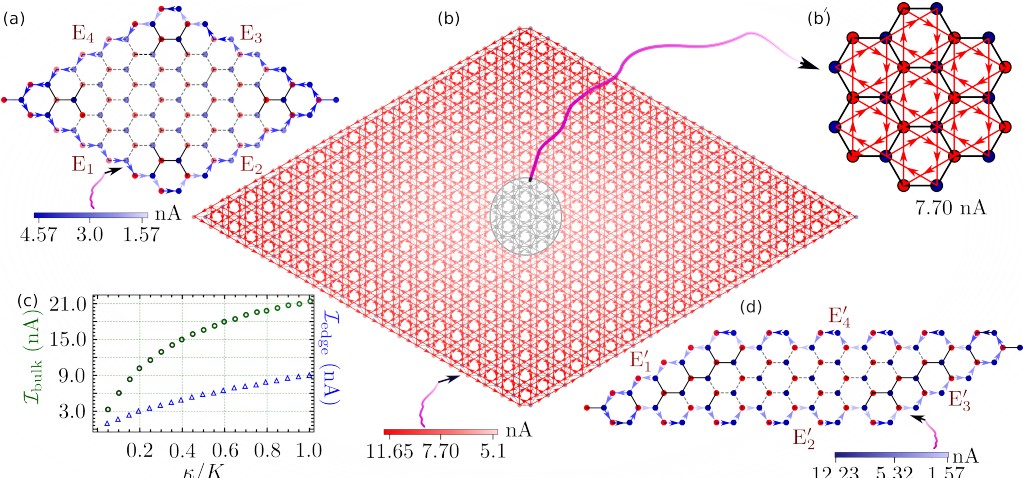

Figure 2: (a) Distribution of the localized edge currents around the zigzag type edges for a finite system with OBC of size $L = 21$ (the bulk of the system is illustrated by the dashed bonds). Note that the localized currents on four different edges are displayed only for the bonds lying on the honeycomb plaquettes. The localized currents on the next-nearest neighbor (NNN) bonds are not shown for simplicity. (b) The macroscopic build-up of localized currents along the NNN bonds is distributed over the entire sample size and forms *filament-like* current textures. A magnified view of the details of the current textures in the middle of the system is shown in the top right panel (b′). (c) The variation of the localized current as a function of $\kappa$ for the nearest-neighbor (NN) bonds on the middle of the edges $E_1 - E_4$ (*open triangles*) and NNN bonds inside the lattice (*open circles*), respectively. (d) The current profile around a different geometry with both the zigzag and arm-chair edges. The numerical estimates are provided with $\kappa = 0.2K$.

When the magnetic field strength becomes large, the perturbative treatment is inadequate. In addition to the $\kappa$ term, the magnetic field generates dynamics for the visons, which spoil the integrability of the model. To make analytical progress, we take $\kappa$ as a free parameter with the understanding that it relates to the magnetic field for a weak field. The main results of the orbital magnetization obtained below are expected to hold even when visons are dynamical. For a large magnetic field, the chiral Kitaev QSL is destroyed and the description in Eq. (7) is no longer applicable [61, 62].

So far we focused on a specific triangular plaquette, however, in a lattice geometry, the situation becomes more interesting. A specific triangle is shared between three honeycomb plaquettes, whereas each nearest-neighbor (NN) bond on the triangle is shared between four triangles. Among these four triangles, two belong to the hexagonal plaquette containing the next-nearest neighbor (NNN) bond $\langle ij \rangle$, where the current operator is defined, and the other two belong to the neighboring hexagonal plaquettes (see Fig. 1). Consequently, the net current along the NN bonds forming the honeycomb plaquettes vanishes because of counter-propagating currents from the two honeycomb plaquettes for translationally invariant systems. In a finite size system with open boundary conditions (OBC), however, such perfect cancellations are not possible near the edge terminations, as shown in Fig. 2. On the other hand, since the NNN bonds are not shared between any plaquettes, the localized currents are never canceled and they create an interesting loop current profile distributed over the entire system, as will be discussed in the next section.

# 5 Finite system analysis

We now focus our analysis on a finite system to compute the localized current profile. We first consider a finite system (with OBC) of linear size $L$ without any $\mathbb{Z}_2$ vortices or vison excitations. We set the value of $\kappa$ motivated by the recent experimental work on $\alpha$-RuCl$_3$ [33]. Comparing the gap magnitude from Eq. (7) with the observed field-dependent Majorana gap at an applied magnetic field of $\sim$ 10 T, we estimate $\kappa \sim 0.77K$. However, this value of $\kappa$ corresponds to a large magnetic field, when our approximate Hamiltonian in Eq. (7) is no longer applicable. Hence, we use a smaller magnitude of $\kappa = 0.2K$ for subsequent quantitative estimates, unless otherwise mentioned. As explained previously, the localized current on a particular bond $\langle ij \rangle_\alpha$ on the honeycomb plaquette is the sum of the contributions from the associated triangles. Following the analytical form in Eq. (6a), it appears as if there are three different average currents $\mathcal{I}_\alpha$ in three different bonds, $\alpha = x, y, z$, respectively [see Fig. 1(a)]. However, the $C_3$ rotation symmetry of the underlying lattice implies that all these three currents are identical i.e., $\mathcal{I}_\alpha = \mathcal{I}$. In addition, neighboring honeycomb plaquettes of a particular bond host counter-propagating current $\mathcal{I}$, leading to vanishing average current distribution at the sides of each honeycomb plaquette in the bulk. However, such perfect cancellations are absent near the edge terminations of the finite system, which consequently leads to non-vanishing localized currents around the edge as shown in Fig. 2(a) and 2(c). The lack of translational invariance around the edges of the system leads to different localized currents along different bonds in each of the triangular plaquettes along the edge. The total current is conserved at each vertex.

On the other hand, the localized currents in each of the NNN bonds $\langle ij \rangle$ within the triangle $\triangle_{ijk}$ do not vanish as they are not shared between other triangles in the lattice geometry. It leads to a macroscopic build-up of the localized currents which mimics a *filament* like structure distributed over the entire system, as shown in Fig. 2(b). For an infinite system, these localized currents on the NNN bonds, form an emergent superlattice of two interpenetrating triangular lattices as illustrated in Fig. 2(b′). We emphasize that there is no free-flowing current over long distances because of the Mott nature of the system. Currents only flow around a closed loop of atomic size. The seemingly free flow of current (red line in Fig. 2(b)) is an illusion due to the superposition of localized current loops. The dependence of the current amplitude as a function of $\kappa$ is shown in Fig. 2(c). In the subsequent sections Sec. 5.1, and 5.2, we focus on two types of edge-termination geometry for finite systems with OBC in the absence of vison excitations. In Sec. 5.3, we again consider a finite system with PBC in the presence of two well-separated $\mathbb{Z}_2$ vortices, and illustrate the drastic modification of the bulk loop currents around these vison excitations.

## 5.1 Quantitative estimates and analysis – case of zigzag geometry

We performed numerical diagonalization for a system of linear size $L = 21$ with zigzag edge terminations to obtain the average localized current distribution without any vison excitations. In this case, the thermodynamic limit is reached for a relative small system size $L \geq 20$ because the system is fully gapped. (see Appendix E for the details). The corresponding results are shown in Fig. 2(a-c). In this geometry, the four different zigzag edges $E_1$ – $E_4$ are related to each other by the three-fold rotation along the $c$-axis, and we focus on one edge viz. $E_1$ for the quantitative analysis.

As shown in Fig. 2(a), the edge $E_1$ is formed by two types of bonds: $x$- and $z$-bond. In the thermodynamic limit, we obtain a localized current that saturates at $\mathcal{I}_{\text{edge}}^{(0)} \sim 3.0$ nA for both the bonds, that lie far away from the corner sites with $E_2$ and $E_4$, respectively. The variation of the the thermodynamic edge current $\mathcal{I}_{\text{edge}}^{(0)}$ as a function of $\kappa$ is shown in Fig. 2(c) (see the variation with open triangles).

The value of $\mathcal{I}_{\text{edge}}$ deviates toward the corners of the system and differs for each bond type because of the lack of translation invariance along the edge. The amplitude for $z(x)$-bond near the corner with $E_4$ is ~ 4.57 (3.83) nA, whereas near the other corner with $E_2$ it is slightly less ~ 2.21 (1.58) nA as shown in Fig. 2(a). This variation along the edge is illustrated by the color legend below panel (a) in Fig. 2. The saturation length (the distance beyond which the bond currents become $\mathcal{I}_{\text{edge}}^{(0)}$) near the edge with $E_4$ is slightly larger than the saturation length near the edge with $E_2$. Furthermore, the currents on the NN bonds inside the system almost vanish as mentioned earlier.

On the other hand, the current amplitude on the NNN bonds [$x', y', z'$-type corresponding to $t_2'$ hopping, see Fig. 1(b)] inside the lattice far away from the edges and corners saturates at $\mathcal{I}_{\text{bulk}}^{(0)}$ ~ 7.70 nA as shown in Fig. 2(b). In comparison to the edge current $\mathcal{I}_{\text{edge}}^{(0)}$, the variation of this thermodynamic bulk localized current $\mathcal{I}_{\text{bulk}}^{(0)}$ as a function of $\kappa$ is shown in Fig. 2(c) (see the variation with open circles). This value also deviates from the bulk value as the bonds come close to the edges/corners of the system. The $C_{3c}$ rotation symmetry, which enforces identical localized currents $\mathcal{I}_{\text{bulk}}^{(0)}$ on all $x', y', z'$ bonds in the bulk, is lost as we go away from the middle of the system to its edge. Consequently, the amplitudes of the localized currents for the three bonds gradually differ as they come closer to the edge/corner as illustrated by the color legend below panel (b) in Fig. 2.

## 5.2 Quantitative estimates and analysis – case of armchair geometry

In the previous section, we focused our quantitative analysis on a finite-size honeycomb lattice with zigzag edges as shown in Fig. 2(a). However, such edge terminations are not the only possible options. Motivated by the graphene literature [63–65], we consider a different geometry with both the zigzag and the armchair edge terminations, as shown in Fig. 2(d). Such geometry can be created by modifying the choice of the unit vectors [66] compared to the previous case. We again perform numerical diagonalization on this geometry of linear size $L = 21$ to obtain the current distribution, without any vison excitations. The localized current on the NNN bonds in the middle of the system saturates at the same value $\mathcal{I}_{\text{bulk}}^{(0)}$ as before. There are four edges and two pairs of edges are related by the inversion operation. Consequently, we focus only on two different edges $E_1'$ and $E_2'$ with zigzag and armchair termination, respectively. For $E_1'$ (with $z$ and $y$-bonds), the current far away from the corners saturates at the same amplitude $\mathcal{I}_{\text{edge}}^{(0)}$ as quoted in the previous section. Of course, this number varies as the bonds come close to the corners as before. This variation is similar to Fig. 2(a).

However, the localized current distribution along the armchair edge $E_2'$ (with all the three bonds – $x, y$ and $z$) is very different from its zigzag counterpart. The amplitude of the current on the outer lying $z$-bonds in the middle of the edge, away from the corners, saturates at a larger value $\mathcal{I}_{\text{z-ach}}^{(0)}$ ~ 10.35 nA, whereas the current on the accompanying $x$ and $y$-bonds saturates at a lower value $\mathcal{I}_{\text{xy-ach}}^{(0)}$ ~ 4.31 nA. The variation of the localized currents on the bonds as they come close to the corners for both the zigzag and the armchair edge terminations is illustrated in Fig. 2(d). The current profile on the NNN bonds is similar to the previous case and is not shown for simplicity. A more interesting edge current profile can be obtained by modifying the edge termination patterns to others such as the bearded edge ribbons [66].

## 5.3 Quantitative estimates and analysis – case of two vison excitations

In this section, we analyze the current distributions in the presence of localized vison excitations. For this purpose, we now consider a finite system with PBC and a bigger linear size $L = 32$ compared to the previous case. We chose a bigger system size to reduce the mutual interaction between the two vison excitations.

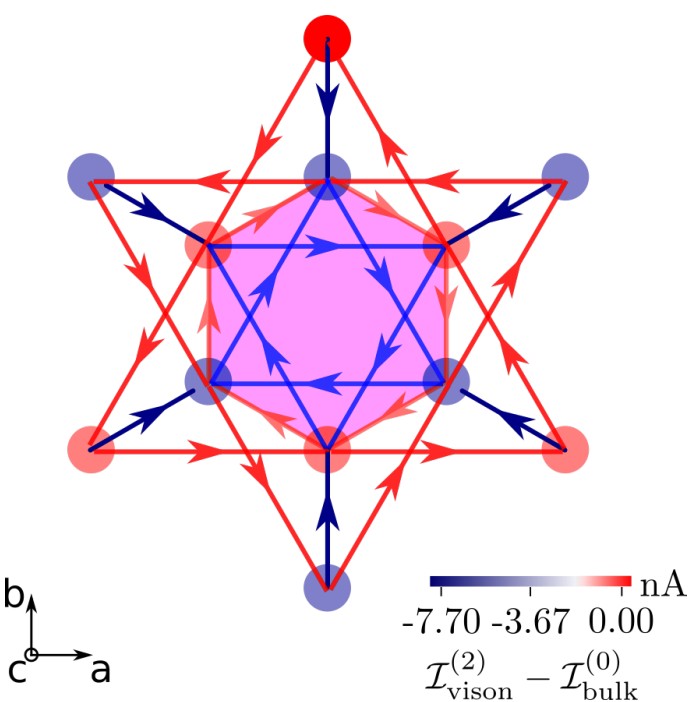

Figure 3: Current profile around a vison excitation (colored hexagon). In contrast to the previous case in Fig. 2, the localized currents on the NN bonds around honeycomb plaquettes containing the vison are nonzero.

We introduce two visons and keep them at the largest possible distance $\lfloor \frac{L-1}{2} \rfloor$ to minimize their mutual interaction effects. Such configuration can be achieved by flipping all the link variables on the $z$-bonds which are connected by the string operator between them, as illustrated in Fig. 1(a). However, as the string operator is gauge-dependent, the specific choice of the string operator connecting two vison excitations does not affect the electrical currents.

The ground state of the *two-vison* Kitaev model, in the presence of an external magnetic field can be similarly obtained, utilizing the Majorana and gauge degrees of freedom, as $|\Psi_{\mathrm{mag}}^{(2)}\rangle = \mathcal{Z}|\mathcal{G}_2\rangle \otimes |\mathcal{M}_{\mathrm{mag}}\rangle$. Here $|\mathcal{G}_2\rangle$ corresponds to the gauge configuration with two visons, and $\mathcal{Z}$ is the projection operator as defined in Sec. (4). We note that the action of the projection operator $\mathcal{Z}$ translates into satisfying a parity constraint for the matter and bond fermions as $(-1)^{n_f + n_\chi} = 1$ (see Appendix E for the technical details) [59, 60]. Next, we compute the average localized current in the ground state $|\Psi_{\mathrm{mag}}^{(2)}\rangle$, by numerically diagonalizing the tight-binding Majorana Hamiltonian with $|\mathcal{G}_2\rangle$ (see Appendix E).

In the previous section with the gauge configuration $|\mathcal{G}_0\rangle$, we observed that the localized currents on the NNN bonds, within the bulk, saturated at a uniform value $\mathcal{I}_{\mathrm{bulk}}^{(0)}$. Such current configuration led to vanishing localized currents on the NN bonds forming the hexagonal plaquettes. In the case with the gauge configuration $|\mathcal{G}_2\rangle$, the localized current profile far away from the static vison excitations is identical to $\mathcal{I}_{\mathrm{bulk}}^{(0)}$.

However, this distribution acquires drastic modification inside and around the vison excitations. The localized currents on the three NNN bonds ($x'$, $y'$, and $z'$-bonds) within the hexagon containing a vison excitation become isotropic but differ from the localized currents on the NNN bonds surrounding the honeycomb plaquettes, and consequently, finite non-vanishing currents emerge on the NN bonds forming the hexagon, as shown in Fig. 3. The localized current on the NNN bonds inside the vison $\mathcal{I}_{\mathrm{vison}}^{(2)}$ saturate at a smaller value of $\sim 4.03$ nA, whereas the corresponding currents on the surrounding hexagons saturates at almost $\mathcal{I}_{\mathrm{bulk}}^{(0)} \sim 7.70$ nA.

Consequently, NN bonds surrounding the hexagon containing the vison excitation host non-zero localized loop currents $\mathcal{I}_{\text{vison-edge}}^{(2)} \sim 7.29$ nA, which is obtained by adding the contributions of the four triangular plaquettes containing the NN bonds. The variation of the amplitudes of these localized currents on the vison plaquettes as a function of the orientation and magnitude of the external magnetic field follows similar dependence as was illustrated in Fig. 2(c) in Sec. (5).

# 6 Loop currents: Implications

So far, we discussed patterns of the TRS breaking localized loop currents in a Kitaev system in the presence of an external magnetic field. For an infinite system, we noticed that the localized loop currents constitute a superlattice formed by inter-penetrating triangular lattices as shown in Fig. 2(b′). The direction of the loop currents is determined by the sign of $\kappa$ and therefore depends on the direction of the external magnetic field. Therefore, in the absence of any vison excitations, one should expect them to contribute a uniform local magnetic moment $\mu_{\text{loop}}$ for each of the hexagonal plaquettes. Since the current orientation for both the triangular plaquettes is the same, the magnetic moment of each of the two triangles adds up and leads to a total orbital magnetic moment in each hexagon as $\mu_{\text{hexagon}} = 2 \times \mu_{\text{loop}}$. Adopting realistic atomic lattice constant for $\alpha$-RuCl$_3$ from the recent *ab intio* study [44], we obtain $\mu_{\text{hexagon}} \sim 0.0003\mu_{\text{B}}$, where $\mu_{\text{B}}$ is the Bohr-magneton. Consequently, we find a weak emergent orbital ferromagnetic magnetic order within the Kitaev QSL phase. Naturally, the vison excitations would create a local change in this emergent long-range orbital ferromagnetic phase as the localized currents are different around the vison excitations. Adopting the previous parameters, we estimate that the magnitude of the local magnetization on an isolated vison to be $\sim \pm 20\%$ of the $\mu_{\text{hexagon}}$. Such orbital ferromagnetic order can be resolved by polarized neutron diffraction [67, 68], muon spin spectroscopy [69], second harmonic generation [70], optical birefringence measurements [71], or superconducting quantum interference device magnetometers [72].

The key feature of the Kitaev system is that for an external magnetic field only applied in $ab$-plane would lead to an induced out-of-plane (along $c$-axis) magnetization. The direction and the magnitude of such $c$-axis magnetic field can be tuned by rotating the magnetic field in the $ab$-plane. Note that the direction of the orbital magnetic field depends on the sign of $\kappa$ which is dictated by Eq. (8).

The magnitude of the localized moments is weak based on our quantitative estimate for $\alpha$-RuCl$_3$. However, two polymorphs of $\alpha$-RuCl$_3$, by replacing the ligand Cl$^-$-ion with Br$^-$ and I$^-$, have recently become possible candidate materials for realizing Kitaev QSL phase [73]. Among these two polymorphs, RuI$_3$ is still debated to be a metal, although RuBr$_3$ is an insulator with almost half the Mott gap compared to RuCl$_3$ [73]. On the other hand, it has a much larger Ru-Ru distance and considerable Ru 4$d$ and Br 4$p$ orbital overlap [74]. Adopting the parameters from the *ab initio* [75] results for RuBr$_3$, we predict an order of magnitude enhancement of such localized magnetic moments for an analogous situation in RuBr$_3$ with $\mu_{\text{hexagon}} \sim 0.003\mu_{\text{B}}$ at $\kappa \sim 0.14K$ if the Kitaev quantum spin liquid phase can be realized there. On the other hand, if we adopt an inflated magnitude of $\kappa$ for RuBr$_3$ motivated by the experimentally measured Majorana gap for $\alpha$-RuCl$_3$ [33] as $\kappa \sim 0.77$, the orbital magnetization can go up as much as $0.01\mu_{\text{B}}$.

# 7 Discussion and conclusion

In this paper, we developed a theoretical framework for analyzing the remnant electrical current responses in spin-orbit coupled multiorbital Mott insulators, such as iridates or ruthenates, considered to be plausible candidate materials to realize the Kitaev QSL phase. Our analysis and predictions reveal an exciting and alternative detection protocol for the signatures of the fractionalized excitations in a Kitaev QSL through emergent orbital magnetization due to the localized orbital currents. This formalism can be generalized to other QSL hosting Mott insulators with broken time-reversal symmetry. Starting from the microscopic multiorbital Hubbard-Kanamori model, we derived the functional form of the induced localized currents in a triangular loop [see Eq. (6a)] using SWT. The current operator expressed in terms of the spin operator in Mott insulators without SU(2) spin rotation symmetry is an important result, on its own, and forms the basis for the rest of our paper. In the presence of an external magnetic field, we solve the Kitaev model in the Majorana representation and notice that the ground state hosts a non-zero expectation value for the localized loop currents. Because of the underlying honeycomb lattice structure, the average non-zero triangular loop current leads to a rich current profile in the two-dimensional system, as shown in Fig. 2. The current on the NNN bonds on the hexagonal plaquettes extends over the entire system and forms a super-lattice of two inter-penetrating triangular lattices as illustrated in Fig. 2(b′), whereas the current on the NN bonds vanishes identically within the bulk. However, depending on the edge termination profiles (zigzag, armchair, or bearded) of the 2D Kitaev system, the localized currents on the NN bonds along the edge can survive and lead to rich and interesting patterns.

We then revisited the analysis of our current profile in the presence of vison excitations. While the localized current distribution in the absence of any visons (uniform gauge configuration) remains identical on the NNN bonds within the bulk, leading to the vanishing currents on NN bonds, a single vison excitation leads to a drastic modification of the currents on the bonds containing the $\mathbb{Z}_2$ vortex (see Fig. 3). Depending on the orientation of the applied magnetic field, the currents on NNN bonds in hexagons containing the $\mathbb{Z}_2$ vortex decrease compared to the surrounding hexagons away from the vison excitation. This leads to a reappearance of non-zero localized currents on the NN bonds around the hexagon with vison excitation, and implies that the visons carry the physical magnetic flux, which makes visons resemble the Abrikosov vortices in superconductors. Note that the inevitable presence of local inhomogeneities in real materials would also leads to modifications of the magnetization profile because of the breakdown of translational invariance. The current pattern induced by impurities can be random and is different from the current pattern induced by a vison. Furthermore, visons are dynamical excitations of the quantum spin liquid, while the orbital current induced by impurities is static. These two distinct features for current induced by impurities and visons can be easily distinguished.

The visons can couple to an external magnetic field directly and can interact through electromagnetic coupling. This direct electromagnetic coupling is generally weak because of the small magnetic moment associated with visons. Under the circumstances when the vison gap is small, one may induce vison lattice by an external magnetic field similar to the Abrikosov vortex lattice.

## Acknowledgments

We would like to thank Philipp Gegenwart, Alexander V. Balatsky, Cristian Batista, Dieter Vollhardt, Krishnendu Sengupta, Ross David Mcdonald, Vivien Zapf, Avadh Saxena, and Umesh

Kumar for useful discussions and careful remarks on the manuscript. This work was carried out under the auspices of the US DOE NNSA under Contract No. 89233218CNA000001 through the LDRD Program, and was performed, in part, at the Center for Integrated Nanotechnologies, an Office of Science User Facility operated for the U.S. DOE Office of Science, under user proposals #2018BU0010 and #2018BU0083.

# A  Schrieffer-Wolff transformation

In this section, we lay out the details of the Schrieffer-Wolff transformation (SWT) to derive the low-energy effective Hamiltonian from a generic strongly correlated electronic model [23,76]. The original correlated Hamiltonian is written as

$$\mathcal{H} = \mathcal{H}_0 + \mathcal{H}_1 \,, \tag{A.1}$$

where $\mathcal{H}_0$ denotes the correlated part of the model (with parameters such as onsite Coulomb repulsion $U$, Hund's coupling $J_{\mathrm{H}}$, etc.), and $\mathcal{H}_1$ denotes the tight-binding (TB) contribution (with parameters including all the hopping parameters $t_1$, $t_2$ or spin-orbit coupling, etc.). Here, various $t_i$'s correspond to the hopping amplitudes between multiple orbitals or sites. In a strongly correlated system, interaction strengths are naturally much larger than the hopping parameters. In this case, we can reduce the full Hamiltonian in Eq. (A.1) to a low-energy effective description to explain various properties of the parent system. SWT is a very powerful tool in this regard, which is achieved via a unitary transformation $U = e^{i\mathcal{S}}$, where $\mathcal{S}$ is a hermitian operator. After applying such a unitary transformation, we obtain the rotated Hamiltonian as

$$
\begin{aligned}
\mathcal{H}' = U^\dagger \mathcal{H} U &= e^{i\mathcal{S}} \mathcal{H} e^{-i\mathcal{S}} \\
&= \mathcal{H} + \left[i\mathcal{S}, \mathcal{H}\right] + \frac{1}{2!}\left[i\mathcal{S},\left[i\mathcal{S},\mathcal{H}\right]\right] + \frac{1}{3!}\left[i\mathcal{S},\left[i\mathcal{S},\left[i\mathcal{S},\mathcal{H}\right]\right]\right] + \cdots \\
&= \mathcal{H}_0 + \mathcal{H}_1 + i\left[\mathcal{S},\mathcal{H}_0\right] + i\left[\mathcal{S},\mathcal{H}_1\right] - \frac{1}{2}\left[\mathcal{S},\left[\mathcal{S},\mathcal{H}_0\right]\right] - \frac{1}{2}\left[\mathcal{S},\left[\mathcal{S},\mathcal{H}_1\right]\right] \\
&\quad - \frac{i}{3!}\left[\mathcal{S},\left[\mathcal{S},\left[\mathcal{S},\mathcal{H}_0\right]\right]\right] - \frac{i}{3!}\left[\mathcal{S},\left[\mathcal{S},\left[\mathcal{S},\mathcal{H}_1\right]\right]\right] + \cdots ,
\end{aligned}
\tag{A.2}
$$

where $\mathcal{S}$ is to be determined later. The idea is now to evaluate the generating function $\mathcal{S}$ in a perturbative manner in terms of the small TB parameter, such that an effective Hamiltonian in each order acts on a truncated Hilbert space avoiding any doubly occupied or empty sites (which corresponds to high-energy configurations). Note that we consider half-filling. For this purpose, we introduce two projection operators $\mathcal{P}$, and $\mathcal{Q}$, where $\mathcal{P}$ projects to singly occupied sites and $\mathcal{Q} = 1 - \mathcal{P}$, correspondingly, projects to either a doubly occupied or an empty site. We now write the generating function $\mathcal{S}$ as a perturbative expansion in terms of the TB parameters as

$$\mathcal{S} = \mathcal{S}^{(1)} + \mathcal{S}^{(2)} + \mathcal{S}^{(3)} + \mathcal{S}^{(4)} + \cdots \tag{A.3}$$

Next, collecting terms of the same order in perturbation, we obtain

$$\mathcal{H}' = \mathcal{H}_0 + \mathcal{H}_1 + i\left[\mathcal{S}^{(1)},\mathcal{H}_0\right] \tag{A.4a}$$

$$+ i\left[\mathcal{S}^{(1)},\mathcal{H}_1\right] + i\left[\mathcal{S}^{(2)},\mathcal{H}_0\right] - \frac{1}{2}\left[\mathcal{S}^{(1)},\left[\mathcal{S}^{(1)}(t),\mathcal{H}_0\right]\right] \tag{A.4b}$$

$$+ i\left[\mathcal{S}^{(3)},\mathcal{H}_0\right] + i\left[\mathcal{S}^{(2)},\mathcal{H}_1\right] - \frac{1}{2}\left[\mathcal{S}^{(1)},\left[\mathcal{S}^{(1)},\mathcal{H}_1\right] + \left[\mathcal{S}^{(2)},\mathcal{H}_0\right]\right]$$

$$- \frac{1}{2}\left[\mathcal{S}^{(2)},\left[\mathcal{S}^{(1)},\mathcal{H}_0\right]\right] - \frac{i}{3!}\left[\mathcal{S}^{(1)},\left[\mathcal{S}^{(1)},\left[\mathcal{S}^{(1)},\mathcal{H}_0\right]\right]\right] \tag{A.4c}$$

$$+ \mathcal{O}(4) \text{ terms} \,, \tag{A.4d}$$

where, in each line, we arranged terms of the same order in perturbation. The rotated Hamiltonian in Eq. (A.4a-A.4d) can be written in a compact form as follows

$$\mathcal{H}' = \sum_{m=0}^{n} \mathcal{H}_{\text{eff}}^{(m)} + \mathcal{O}(n+1), \tag{A.5}$$

where the effective Hamiltonian in $m$-th perturbative order $\mathcal{H}_{\text{eff}}^{(m)}$ is computed in such a way that it does not have any mixing terms, *i.e.*, $\mathcal{P}\mathcal{H}_{\text{eff}}^{(m)}\mathcal{Q} = \mathcal{Q}\mathcal{H}_{\text{eff}}^{(m)}\mathcal{P} = 0$. In the subsequent analysis, we provide the key steps to obtain the analytical structure of the effective Hamiltonian up to third-order in the perturbation expansion.

## A.1 Second-order effective Hamiltonian

We now move on to the analysis of the second-order effective Hamiltonian and obtain the equation for the generating function $\mathcal{S}^{(1)}$ from Eq. (A.4a). Utilizing this relation in Eq. (A.4b), we obtain the effective Hamiltonian in the second-order perturbation as

$$\left[\mathcal{S}^{(1)}, \mathcal{H}_0\right] = i\mathcal{H}_1, \tag{A.6a}$$

$$\mathcal{H}_{\text{eff}}^{(2)} = \frac{i}{2}\left[\mathcal{S}^{(1)}, \mathcal{H}_1\right]. \tag{A.6b}$$

To solve for the generating function $\mathcal{S}^{(1)}$, we first notice its $2 \times 2$ matrix structure in the basis of the projection operators $\mathcal{P}$, and $\mathcal{Q}$ as

$$\mathcal{S}^{(1)} = \begin{pmatrix} \mathcal{P}\mathcal{S}^{(1)}\mathcal{P} & \mathcal{P}\mathcal{S}^{(1)}\mathcal{Q} \\ \mathcal{Q}\mathcal{S}^{(1)}\mathcal{P} & \mathcal{Q}\mathcal{S}^{(1)}\mathcal{Q} \end{pmatrix}, \tag{A.7}$$

where $\mathcal{P}$, and $\mathcal{Q}$ are defined earlier. Since, we need to remove the off-diagonal elements, the diagonal elements $[\mathcal{P}\mathcal{S}^{(1)}\mathcal{P}, \mathcal{Q}\mathcal{S}^{(1)}\mathcal{Q}]$ are naturally assumed to be zero, and the off-diagonal elements are obtained from Eq. (A.6a) as

$$\mathcal{P}\mathcal{S}^{(1)}\mathcal{H}_0\mathcal{Q} - \mathcal{P}\mathcal{H}_0\mathcal{S}^{(1)}\mathcal{Q} = i\mathcal{P}\mathcal{H}_1\mathcal{Q}$$
$$\Rightarrow \mathcal{P}\mathcal{S}^{(1)}\mathcal{Q}\mathcal{Q}\mathcal{H}_0\mathcal{Q} - \cancel{\mathcal{P}\mathcal{H}_0\mathcal{Q}\mathcal{Q}\mathcal{S}^{(1)}\mathcal{Q}} = i\mathcal{P}\mathcal{H}_1\mathcal{Q}, \tag{A.8a}$$

$$\mathcal{Q}\mathcal{S}^{(1)}\mathcal{H}_0\mathcal{P} - \mathcal{Q}\mathcal{H}_0\mathcal{S}^{(1)}\mathcal{P} = i\mathcal{Q}\mathcal{H}_1\mathcal{P}$$
$$\Rightarrow \cancel{\mathcal{Q}\mathcal{S}^{(1)}(t)\mathcal{Q}\mathcal{Q}\mathcal{H}_0\mathcal{P}} - \mathcal{Q}\mathcal{H}_0\mathcal{Q}\mathcal{Q}\mathcal{S}^{(1)}\mathcal{P} = i\mathcal{Q}\mathcal{H}_1\mathcal{P}. \tag{A.8b}$$

## A.2 Third-order effective Hamiltonian

The third-order effective Hamiltonian can be obtained similarly as outlined in Sec. (A.1). We first obtain the equation for the generating function $\mathcal{S}^{(2)}$ [see Eq. (A.4b)], and utilize it to derive the effective Hamiltonian in the third-order perturbation. Consequently, we obtain [Note: the last term in Eq. (A.4b) can be recast in the form as in Eq. (A.6b) by utilizing the equation of motion for $\mathcal{S}^{(1)}$ in Eq. (A.6a)]

$$\left[\mathcal{S}^{(2)}, \mathcal{H}_0\right] = -\left[\mathcal{S}^{(1)}, \mathcal{H}_1\right] \tag{A.9a}$$

$$\mathcal{H}_{\text{eff}}^{(3)} = \frac{i}{2}\left[\mathcal{S}^{(2)}, \mathcal{H}_1\right] + \frac{1}{6}\left[\mathcal{S}^{(1)}, \left[\mathcal{S}^{(1)}, \mathcal{H}_1\right]\right]. \tag{A.9b}$$

Finally, we evaluate the matrix elements for $\mathcal{S}^{(2)}$ in the projection operator basis as

$$\mathcal{P}\mathcal{S}^{(2)}\mathcal{Q}\mathcal{Q}\mathcal{H}_0\mathcal{Q} - \cancel{\mathcal{P}\mathcal{H}_0\mathcal{Q}\mathcal{Q}\mathcal{S}^{(2)}\mathcal{Q}} = \cancel{\mathcal{P}\mathcal{H}_1\mathcal{Q}\mathcal{Q}\mathcal{S}^{(1)}\mathcal{Q}} - \mathcal{P}\mathcal{S}^{(1)}\mathcal{Q}\mathcal{Q}\mathcal{H}_1\mathcal{Q}$$
$$\Rightarrow \mathcal{P}\mathcal{S}^{(2)}\mathcal{Q}\mathcal{Q}\mathcal{H}_0\mathcal{Q} = -\mathcal{P}\mathcal{S}^{(1)}\mathcal{Q}\mathcal{Q}\mathcal{H}_1\mathcal{Q}, \tag{A.10a}$$

$$\cancel{\mathcal{Q}\mathcal{S}^{(2)}\mathcal{Q}\mathcal{Q}\mathcal{H}_0\mathcal{P}} - \mathcal{Q}\mathcal{H}_0\mathcal{Q}\mathcal{Q}\mathcal{S}^{(2)}\mathcal{P} = \mathcal{Q}\mathcal{H}_1\mathcal{Q}\mathcal{Q}\mathcal{S}^{(1)}\mathcal{P} - \cancel{\mathcal{Q}\mathcal{S}^{(1)}\mathcal{Q}\mathcal{Q}\mathcal{H}_1\mathcal{P}}$$
$$\Rightarrow \mathcal{Q}\mathcal{H}_0\mathcal{Q}\mathcal{Q}\mathcal{S}^{(2)}\mathcal{P} = -\mathcal{Q}\mathcal{H}_1\mathcal{Q}\mathcal{Q}\mathcal{S}^{(1)}\mathcal{P}, \tag{A.10b}$$

where $\mathcal{Q}\mathcal{H}_1\mathcal{Q}$ corresponds to hopping between either two doubly occupied states or two singly occupied sites (always leaving one empty site after the hopping, in the latter case).

# B  Effective analytical structure of a generic operator

In previous Appendix (A), we provided all the details of the SWT to obtain the generic forms of the effective low-energy Hamiltonian. Once the generating functions $\mathcal{S}^{(m)}$ are obtained upto $m$-th order in perturbation, any other operators can be rotated in the same manner as in Eq. (A.2). Consequently, in the rotated frame, any local operator $\mathcal{O}_i$ becomes

$$\tilde{\mathcal{O}}_i = e^{i\mathcal{S}}\mathcal{O}_i e^{-i\mathcal{S}} = \mathcal{O}_i + i[\mathcal{S}, \mathcal{O}_i] - \frac{1}{2!}[\mathcal{S}, [\mathcal{S}, \mathcal{O}_i]] - \frac{i}{3!}[\mathcal{S}, [\mathcal{S}, [\mathcal{S}, \mathcal{O}_i]]] + \cdots, \qquad (B.1)$$

where $\mathcal{O}_i$ a generic physical operator *viz.*, the local charge imbalance operator at half-filling *i.e.* $\delta\boldsymbol{\rho}_i = e\left(d^\dagger_{i\alpha\sigma} d_{i\alpha\sigma} - 1\right)$, or the current operator *i.e.* $\boldsymbol{\mathcal{I}}_{ij} = \frac{iet'\hat{\mathbf{r}}_{ij}}{\hbar}\sum_{\alpha\beta}\left(d^\dagger_{j\alpha\sigma} d_{i\beta\sigma} - d^\dagger_{i\beta\sigma} d_{j\alpha\sigma}\right)$, where the sum over repeated indices are assumed and $(\alpha, \beta)$, and $\sigma$ correspond to the orbital, and spin degrees of freedom, respectively. Utilizing the projection operators $\mathcal{P}$ and $\mathcal{Q}$, as defined earlier, we analyze the low-energy effective structure for $\mathcal{O}_i$ as

$$\mathcal{P}\tilde{\mathcal{O}}_i\mathcal{P} = \cancel{\mathcal{P}\mathcal{O}_i\mathcal{P}} + i\cancel{\mathcal{P}[\mathcal{S}, \mathcal{O}_i]\mathcal{P}} - \frac{1}{2!}\mathcal{P}[\mathcal{S}, [\mathcal{S}, \mathcal{O}_i]]\mathcal{P} - \frac{i}{3!}\mathcal{P}[\mathcal{S}, [\mathcal{S}, [\mathcal{S}, \mathcal{O}_i]]]\mathcal{P} + \cdots$$

$$= -\frac{1}{2!}\mathcal{P}[\mathcal{S}, [\mathcal{S}, \mathcal{O}_i]]\mathcal{P} - \frac{i}{3!}\mathcal{P}[\mathcal{S}, [\mathcal{S}, [\mathcal{S}, \mathcal{O}_i]]]\mathcal{P} + \cdots, \qquad (B.2)$$

where we utilized the relation $\mathcal{P}\mathcal{O}_i\mathcal{P} = 0$ and $\mathcal{P}[\mathcal{S}, \mathcal{O}_i]\mathcal{P} = 0$. This property holds for the case where $\mathcal{O}_i$ is a diagonal operator in the projection operator space. An example for this is the charge imbalance operator $\delta\boldsymbol{\rho}_i$ [39]. However, for the current operator $\boldsymbol{\mathcal{I}}_{ij}$ (off-diagonal in $\mathcal{P}$-$\mathcal{Q}$ space), such a restriction does not hold *i.e.* $\mathcal{P}[\mathcal{S}, \boldsymbol{\mathcal{I}}_{ij}]\mathcal{P} \neq 0$. We now focus our analysis on the low-energy effective form for the current operator.

## B.1  Circulating loop current

Since there are no mobile-charged carriers in a Mott insulator, a free-flowing current cannot exist in the system. However, this constraint does not exclude the possibility of having an induced circulating loop current in a closed loop. A minimal closed loop is made out of three neighboring sites forming a triangular plaquette. Here, we derive the localized loop current operator up to third-order in perturbation. Formally it reads as

$$\tilde{\boldsymbol{\mathcal{I}}}^{(2)}_{ij,k} = i\mathcal{P}[\mathcal{S}^{(2)}, \boldsymbol{\mathcal{I}}_{ij}]\mathcal{P} - \frac{1}{2}\mathcal{P}[\mathcal{S}^{(1)}, [\mathcal{S}^{(1)}, \boldsymbol{\mathcal{I}}_{ij}]]\mathcal{P}$$

$$= i\mathcal{P}\mathcal{S}^{(2)}\mathcal{Q}\mathcal{Q}\boldsymbol{\mathcal{I}}_{ij}\mathcal{P} + \mathcal{P}\mathcal{S}^{(1)}\mathcal{Q}\mathcal{Q}\boldsymbol{\mathcal{I}}_{ij}\mathcal{Q}\mathcal{Q}\mathcal{S}^{(1)}\mathcal{P} + \text{h.c.}, \qquad (B.3)$$

where we utilized the equation for the generating functions $\mathcal{S}^{(1)}$ and $\mathcal{S}^{(2)}$ as in Eqs. (A.8a-A.8b) and Eqs. (A.10a-A.10b). The form of the current operator in a single-orbital Hubbard model [37] can be obtained utilizing the above scheme. Here, we apply this scheme to the multiorbital Hubbard-Kanamori Hamiltonian, relevant for $\alpha$-RuCl$_3$ [77,78], to derive the low-energy form of the current operator.

## B.2 Hubbard-Kanamori model

The microscopic multiorbital Hamiltonian for Kiteav magnets such as $\alpha$-RuCl$_3$ is written as (see Refs. [ [16]],[ [18]])

$$\mathcal{H} = \mathcal{H}_0 + \mathcal{H}_1 + \mathcal{H}_2 \,, \tag{B.4a}$$

$$\mathcal{H}_0 = U \sum_{i\alpha} n_{i\alpha,\uparrow} n_{i\alpha,\downarrow} + \frac{U'}{2} \sum_{\substack{\alpha\neq\beta, \\ \sigma,\sigma'}} n_{i\alpha\sigma} n_{i\beta\sigma'} - \frac{J_H}{2} \sum_{\substack{\alpha\neq\beta, \\ \sigma,\sigma'}} d^\dagger_{i\alpha\sigma} d_{i\alpha\sigma'} d^\dagger_{i\beta\sigma'} d_{i\beta\sigma} + \frac{\lambda}{2} \sum_i d^\dagger_i \left( \mathbf{L}\cdot\mathbf{S} \right) d_i \,, \tag{B.4b}$$

$$\mathcal{H}_1 = \sum_{\langle ij\rangle\sigma} \begin{pmatrix} d^\dagger_{ixz\sigma} & d^\dagger_{iyz\sigma} & d^\dagger_{ixy\sigma} \end{pmatrix} \begin{pmatrix} 0 & t_2 & 0 \\ t_2 & 0 & 0 \\ 0 & 0 & 0 \end{pmatrix} \begin{pmatrix} d_{jxz\sigma} \\ d_{jyz\sigma} \\ d_{jxy\sigma} \end{pmatrix},$$

$$\mathcal{H}_2 = \sum_{\langle\!\langle ij\rangle\!\rangle\sigma} \begin{pmatrix} d^\dagger_{ixz\sigma} & d^\dagger_{iyz\sigma} & d^\dagger_{ixy\sigma} \end{pmatrix} \begin{pmatrix} 0 & t'_2 & 0 \\ t'_2 & 0 & 0 \\ 0 & 0 & 0 \end{pmatrix} \begin{pmatrix} d_{jxz\sigma} \\ d_{jyz\sigma} \\ d_{jxy\sigma} \end{pmatrix}, \tag{B.4c}$$

where $(\mathcal{H}_2)$ $\mathcal{H}_1$ in Eq. (B.4c) is the (next)-nearest-neighbor hopping matrix between the $d_{xz}$, $d_{yz}$ and $d_{xy}$ orbitals in the $t_{2g}$ manifold. The specific tight-binding structure of $\mathcal{H}_2$ is adopted from Ref. [ [79]]. For the Kanamori part in Eq. (B.4b) $U$ is the strength of the onsite Coulomb repulsion, $J_H$ is the Hund's coupling, $\lambda$ is the atomic spin-orbit coupling. For rotational invariant system, we consider $U' = U - 2J_H$ [23, 24]. Since each individual hopping between the neighboring magnetic ions is oriented along the particular type of the bond [*see Fig. 1(b) in the main text*], we need to modify the above Slater-Koster TB model depending on the particular planar orientation of the bonds. In Eq. (B.4c), we assumed a generic notation and wrote both the nearest and next-nearest neighbor hopping assuming the corresponding bond to be lying in the $xy$-plane (**z**-bond). However, in the actual crystalline environment, the three individual hopping matrix elements (along three different bonds-**x**, **y**, **z**) are modified as (for the next-neighbor bonds we label the bond types as $\mathbf{x}', \mathbf{y}', \mathbf{z}'$)

$$\mathcal{H}_1 = \sum_{\substack{\langle ij\rangle_{\mathbf{z}} \\ \sigma}} \begin{pmatrix} d^\dagger_{ixz\sigma} & d^\dagger_{iyz\sigma} & d^\dagger_{ixy\sigma} \end{pmatrix} \begin{pmatrix} 0 & t_2 & 0 \\ t_2 & 0 & 0 \\ 0 & 0 & 0 \end{pmatrix} \begin{pmatrix} d_{jxz\sigma} \\ d_{jyz\sigma} \\ d_{jxy\sigma} \end{pmatrix}, \tag{B.5a}$$

$$\mathcal{H}_1 = \sum_{\substack{\langle jk\rangle_{\mathbf{y}} \\ \sigma}} \begin{pmatrix} d^\dagger_{jxz\sigma} & d^\dagger_{jyz\sigma} & d^\dagger_{jxy\sigma} \end{pmatrix} \begin{pmatrix} 0 & 0 & 0 \\ 0 & 0 & t_2 \\ 0 & t_2 & 0 \end{pmatrix} \begin{pmatrix} d_{kxz\sigma} \\ d_{kyz\sigma} \\ d_{kxy\sigma} \end{pmatrix}, \tag{B.5b}$$

$$\mathcal{H}_1 = \sum_{\substack{\langle ki\rangle_{\mathbf{x}} \\ \sigma}} \begin{pmatrix} d^\dagger_{kxz\sigma} & d^\dagger_{kyz\sigma} & d^\dagger_{kxy\sigma} \end{pmatrix} \begin{pmatrix} 0 & 0 & t_2 \\ 0 & 0 & 0 \\ t_2 & 0 & 0 \end{pmatrix} \begin{pmatrix} d_{ixz\sigma} \\ d_{iyz\sigma} \\ d_{ixy\sigma} \end{pmatrix}, \tag{B.5c}$$

$$\mathcal{H}_2 = \mathcal{H}_1 \left[ \langle ij\rangle \to \langle\!\langle ij\rangle\!\rangle, \ \ t_2 \to t'_2, \ \ (\mathbf{x},\mathbf{y},\mathbf{z}) \to (\mathbf{x}',\mathbf{y}',\mathbf{z}') \right], \tag{B.5d}$$

where the subscripts for bond-type are chosen according to Fig. 1(b) in the main text. For subsequent analysis, we adopt all the parameters entering Eqs. (B.4a-B.5d) from the recent *ab initio* [44] and *photoemission studies* [45] for $\alpha$-RuCl$_3$ as: $U = 3.0$ eV, $J_H = 0.45$ eV, $t_2 = 0.191$ eV, and $t'_2 = -0.058$ eV. For subsequent analysis, we first rewrite the Hamiltonian in Eq. (B.4a) in the irreducible representation of the doubly occupied states of the octahedral point group (O$_h$) as

$$\mathcal{H}_0 = \sum_i \sum_\Gamma \sum_{g_\Gamma} U_\Gamma \left| i; \Gamma, g_\Gamma \right\rangle \left\langle i; \Gamma, g_\Gamma \right| \,, \tag{B.6}$$

where $\Gamma$ corresponds to a particular irreducible representation and $g_\Gamma$ characterizes its degeneracy. The energy of the three non-degenerate states are given as follows [23, 80, 81]

$$U_{A_1} = U + 2J_H,\tag{B.7a}$$

$$U_E = U - J_H,\tag{B.7b}$$

$$U_{T_1} = U - 3J_H,\tag{B.7c}$$

$$U_{T_2} = U - J_H.\tag{B.7d}$$

There are three orbitals and two spin degrees of freedom, and we have to put two electrons within this manifold. Hence, there are $^6C_2 = 15$ possibility of doubly occupied states. From the character table of $O_h$, we write these fifteen intermediate doubly occupied states at site $i$ as (spanned by the Hilbert space of the projection operator $\mathcal{Q}$)

$$|i;A_1\rangle = \frac{1}{\sqrt{3}}(d^\dagger_{ixz\uparrow}d^\dagger_{ixz\downarrow} + d^\dagger_{iyz\uparrow}d^\dagger_{iyz\downarrow} + d^\dagger_{ixy\uparrow}d^\dagger_{ixy\downarrow})|0\rangle,\tag{B.8a}$$

$$|i;E,u\rangle = \frac{1}{\sqrt{6}}(d^\dagger_{iyz\uparrow}d^\dagger_{iyz\downarrow} + d^\dagger_{ixz\uparrow}d^\dagger_{ixz\downarrow} - 2d^\dagger_{ixy\uparrow}d^\dagger_{ixy\downarrow})|0\rangle,\tag{B.8b}$$

$$|i;E,v\rangle = \frac{1}{\sqrt{2}}(d^\dagger_{iyz\uparrow}d^\dagger_{iyz\downarrow} - d^\dagger_{ixz\uparrow}d^\dagger_{ixz\downarrow})|0\rangle,\tag{B.8c}$$

$$|i;T_1,\alpha_+\rangle = d^\dagger_{iyz\uparrow}d^\dagger_{izx\uparrow}|0\rangle,\tag{B.8d}$$

$$|i;T_1,\alpha_-\rangle = d^\dagger_{iyz\downarrow}d^\dagger_{izx\downarrow}|0\rangle,\tag{B.8e}$$

$$|i;T_1,\alpha\rangle = \frac{1}{\sqrt{2}}(d^\dagger_{iyz\uparrow}d^\dagger_{izx\downarrow} + d^\dagger_{iyz\downarrow}d^\dagger_{izx\uparrow})|0\rangle,\tag{B.8f}$$

$$|i;T_1,\beta_+\rangle = d^\dagger_{izx\uparrow}d^\dagger_{ixy\uparrow}|0\rangle,\tag{B.8g}$$

$$|i;T_1,\beta_-\rangle = d^\dagger_{izx\downarrow}d^\dagger_{ixy\downarrow}|0\rangle,\tag{B.8h}$$

$$|i;T_1,\beta\rangle = \frac{1}{\sqrt{2}}(d^\dagger_{izx\uparrow}d^\dagger_{ixy\downarrow} + d^\dagger_{izx\downarrow}d^\dagger_{ixy\uparrow})|0\rangle,\tag{B.8i}$$

$$|i;T_1,\gamma_+\rangle = d^\dagger_{ixy\uparrow}d^\dagger_{iyz\uparrow}|0\rangle,\tag{B.8j}$$

$$|i;T_1,\gamma_-\rangle = d^\dagger_{ixy\downarrow}d^\dagger_{iyz\downarrow}|0\rangle,\tag{B.8k}$$

$$|i;T_1,\gamma\rangle = \frac{1}{\sqrt{2}}(d^\dagger_{ixy\uparrow}d^\dagger_{iyz\downarrow} + d^\dagger_{ixy\downarrow}d^\dagger_{iyz\uparrow})|0\rangle,\tag{B.8l}$$

$$|i;T_2,\alpha\rangle = \frac{1}{\sqrt{2}}(d^\dagger_{iyz\uparrow}d^\dagger_{izx\downarrow} - d^\dagger_{iyz\downarrow}d^\dagger_{izx\uparrow})|0\rangle,\tag{B.8m}$$

$$|i;T_2,\beta\rangle = \frac{1}{\sqrt{2}}(d^\dagger_{izx\uparrow}d^\dagger_{ixy\downarrow} - d^\dagger_{izx\downarrow}d^\dagger_{ixy\uparrow})|0\rangle,\tag{B.8n}$$

$$|i;T_2,\gamma\rangle = \frac{1}{\sqrt{2}}(d^\dagger_{ixy\uparrow}d^\dagger_{iyz\downarrow} - d^\dagger_{ixy\downarrow}d^\dagger_{iyz\uparrow})|0\rangle.\tag{B.8o}$$

The singly occupied states at site $i$ are written as (spanned by the Hilbert space of the projection operator $\mathcal{P}$)

$$|i,+\rangle = \frac{1}{\sqrt{3}}\left(id^\dagger_{ixz\downarrow} + d^\dagger_{iyz\downarrow} + d^\dagger_{ixy\uparrow}\right)|0\rangle,\tag{B.9a}$$

$$|i,-\rangle = \frac{1}{\sqrt{3}}\left(id^\dagger_{ixz\uparrow} - d^\dagger_{iyz\uparrow} + d^\dagger_{ixy\downarrow}\right)|0\rangle.\tag{B.9b}$$

Consequently, the low-energy Hilbert space for three site problem is written in terms of the eight states ($2^3$) as (considering a three-site triangular subsystem within the crystal)

$$|+,+,+\rangle = \frac{1}{3\sqrt{3}}\left(id^\dagger_{ixz\downarrow} + d^\dagger_{iyz\downarrow} + d^\dagger_{ixy\uparrow}\right)\left(id^\dagger_{jxz\downarrow} + d^\dagger_{jyz\downarrow} + d^\dagger_{jxy\uparrow}\right)\left(id^\dagger_{kxz\downarrow} + d^\dagger_{kyz\downarrow} + d^\dagger_{kxy\uparrow}\right)|0\rangle\,, \quad \text{(B.10a)}$$

$$|+,+,-\rangle = \frac{1}{3\sqrt{3}}\left(id^\dagger_{ixz\downarrow} + d^\dagger_{iyz\downarrow} + d^\dagger_{ixy\uparrow}\right)\left(id^\dagger_{jxz\downarrow} + d^\dagger_{jyz\downarrow} + d^\dagger_{jxy\downarrow}\right)\left(id^\dagger_{kxz\downarrow} - d^\dagger_{kyz\downarrow} + d^\dagger_{kxy\uparrow}\right)|0\rangle\,, \quad \text{(B.10b)}$$

$$|+,-,+\rangle = \frac{1}{3\sqrt{3}}\left(id^\dagger_{ixz\uparrow} + d^\dagger_{iyz\uparrow} + d^\dagger_{ixy\downarrow}\right)\left(id^\dagger_{jxz\downarrow} - d^\dagger_{jyz\downarrow} + d^\dagger_{jxy\uparrow}\right)\left(id^\dagger_{kxz\downarrow} + d^\dagger_{kyz\downarrow} + d^\dagger_{kxy\uparrow}\right)|0\rangle\,, \quad \text{(B.10c)}$$

$$|+,-,-\rangle = \frac{1}{3\sqrt{3}}\left(id^\dagger_{ixz\uparrow} + d^\dagger_{iyz\uparrow} + d^\dagger_{ixy\downarrow}\right)\left(id^\dagger_{jxz\uparrow} - d^\dagger_{jyz\downarrow} + d^\dagger_{jxy\downarrow}\right)\left(id^\dagger_{kxz\downarrow} - d^\dagger_{kyz\downarrow} + d^\dagger_{kxy\uparrow}\right)|0\rangle\,, \quad \text{(B.10d)}$$

$$|-,+,+\rangle = \frac{1}{3\sqrt{3}}\left(id^\dagger_{ixz\uparrow} - d^\dagger_{iyz\uparrow} + d^\dagger_{ixy\downarrow}\right)\left(id^\dagger_{jxz\downarrow} + d^\dagger_{jyz\downarrow} + d^\dagger_{jxy\downarrow}\right)\left(id^\dagger_{kxz\downarrow} + d^\dagger_{kyz\downarrow} + d^\dagger_{kxy\uparrow}\right)|0\rangle\,, \quad \text{(B.10e)}$$

$$|-,+,-\rangle = \frac{1}{3\sqrt{3}}\left(id^\dagger_{ixz\uparrow} - d^\dagger_{iyz\uparrow} + d^\dagger_{ixy\downarrow}\right)\left(id^\dagger_{jxz\uparrow} + d^\dagger_{jyz\uparrow} + d^\dagger_{jxy\downarrow}\right)\left(id^\dagger_{kxz\downarrow} - d^\dagger_{kyz\downarrow} + d^\dagger_{kxy\uparrow}\right)|0\rangle\,, \quad \text{(B.10f)}$$

$$|-,-,+\rangle = \frac{1}{3\sqrt{3}}\left(id^\dagger_{ixz\uparrow} - d^\dagger_{iyz\uparrow} + d^\dagger_{ixy\downarrow}\right)\left(id^\dagger_{jxz\uparrow} - d^\dagger_{jyz\uparrow} + d^\dagger_{jxy\downarrow}\right)\left(id^\dagger_{kxz\downarrow} + d^\dagger_{kyz\downarrow} + d^\dagger_{kxy\uparrow}\right)|0\rangle\,, \quad \text{(B.10g)}$$

$$|-,-,-\rangle = \frac{1}{3\sqrt{3}}\left(id^\dagger_{ixz\uparrow} - d^\dagger_{iyz\uparrow} + d^\dagger_{ixy\downarrow}\right)\left(id^\dagger_{jxz\uparrow} - d^\dagger_{jyz\uparrow} + d^\dagger_{jxy\downarrow}\right)\left(id^\dagger_{kxz\downarrow} - d^\dagger_{kyz\downarrow} + d^\dagger_{kxy\uparrow}\right)|0\rangle\,. \quad \text{(B.10h)}$$

### B.3 Third-order effective form: Induced loop current operator

In this section, we outline the derivation the induced localized loop current operator in the third-order perturbation expansion in terms of the TB parameters $t_2, t'_2$ using Eq. (B.3). Explicitly writing Eq. (B.3) with the individual hoppings, we obtain the operator expression for the localized loop current as

$$\begin{aligned}
\tilde{\mathcal{I}}^{(2)}_{ij,k} = \frac{\mathcal{I}_0 t_2^2 t'_2}{U_\Gamma U_{\Gamma'}} \sum_{\{\alpha,\sigma\}} \Big[ &\mathcal{P}id^\dagger_{i\alpha\sigma''}d_{k\beta\sigma''}\mathcal{Q}_{\Gamma'}\mathcal{Q}_{\Gamma'}d^\dagger_{k\gamma\sigma'}d_{j\delta\sigma'}\mathcal{Q}_\Gamma\mathcal{Q}_\Gamma d^\dagger_{j\eta\sigma}d_{i\kappa\sigma}\mathcal{P} \\
&+\mathcal{P}id^\dagger_{k\alpha\sigma''}d_{j\beta\sigma''}\mathcal{Q}_{\Gamma'}\mathcal{Q}_{\Gamma'}d^\dagger_{i\gamma\sigma'}d_{k\delta\sigma'}\mathcal{Q}_\Gamma\mathcal{Q}_\Gamma d^\dagger_{j\eta\sigma}d_{i\kappa\sigma}\mathcal{P} \\
&+\mathcal{P}id^\dagger_{j\alpha\sigma''}d_{i\beta\sigma''}\mathcal{Q}_{\Gamma'}\mathcal{Q}_{\Gamma'}d^\dagger_{i\gamma\sigma'}d_{k\delta\sigma'}\mathcal{Q}_\Gamma\mathcal{Q}_\Gamma d^\dagger_{k\kappa\sigma}d_{j\eta\sigma}\mathcal{P} \\
&+\mathcal{P}id^\dagger_{j\alpha\sigma''}d_{i\beta\sigma''}\mathcal{Q}_{\Gamma'}\mathcal{Q}_{\Gamma'}d^\dagger_{k\gamma\sigma'}d_{j\delta\sigma'}\mathcal{Q}_\Gamma\mathcal{Q}_\Gamma d^\dagger_{i\kappa\sigma}d_{k\eta\sigma}\mathcal{P} \\
&+\mathcal{P}id^\dagger_{k\alpha\sigma''}d_{j\beta\sigma''}\mathcal{Q}_\Gamma\mathcal{Q}_\Gamma d^\dagger_{j\gamma\sigma}d_{i\delta\sigma}\mathcal{Q}_{\Gamma'}\mathcal{Q}_{\Gamma'}d^\dagger_{i\eta\sigma'}d_{k\kappa\sigma'}\mathcal{P} \\
&+\mathcal{P}id^\dagger_{i\alpha\sigma''}d_{k\beta\sigma''}\mathcal{Q}_\Gamma\mathcal{Q}_\Gamma d^\dagger_{j\delta\sigma}d_{i\gamma\sigma}\mathcal{Q}_{\Gamma'}\mathcal{Q}_{\Gamma'}d^\dagger_{k\eta\sigma'}d_{j\kappa\sigma'}\mathcal{P} + \text{h.c.} \Big]\,,
\end{aligned} \quad \text{(B.11)}$$

where we have the projection operator $\mathcal{Q}$ to $\mathcal{Q}_\Gamma$ to denote all the fifteen eigenstates as defined in Eq. (B.8a-B.8o) with $U_\Gamma$ being the corresponding eigen-energy [see Eq. (B.7a-B.7d)], and $\mathcal{I}_0 = et'_2\hat{\mathbf{r}}_{ij}/\hbar$ is the amplitude of the current operator defined on the $\mathbf{z}'$ bond [*see Fig. 1(b) in the main text*]. Note that we defined the current operator on the longer $\langle ij\rangle$-bond [see Fig. 1(b) in the main text]. After careful analysis of the six terms in Eq. (B.11), we notice that there are two nonequivalent classes of hopping processes: (a) intermediate states are two distinct doublets at two different sites, and (b) intermediate state is a single doublet at the same site. Considering our geometry [*see Fig. 1(b) in the main text*], we can write these processes (a), (b) as:

- (a) $i \to j \Rightarrow j \to k \Rightarrow k \to i$, and $j \to k \Rightarrow k \to i \Rightarrow i \to j$ and $k \to i \Rightarrow i \to j \Rightarrow j \to k$,

- (b) $i \to j \Rightarrow \underline{k \to i} \Rightarrow j \to k$, and $j \to k \Rightarrow \underline{i \to j} \Rightarrow k \to i$ and $k \to i \Rightarrow \underline{j \to k} \Rightarrow i \to j$.

We further notice that in the process (b) we always have one hopping process that connects two singly occupied sites. As the initial and final configuration are constrained by the eight states with $J_{\text{eff}} = 1/2$ total angular momentum [*see Eqs. (B.10a-B.10h)*], the overlap amplitude between such singly occupied states with $t_2/t_2'$ [*underlined processes in (b)*] hopping always vanishes. Hence, the only contribution to the induced current operator comes from the three processes listed in item [(a)]. We use DiracQ package in Mathematica [82] to compute the matrix elements in Eq. (B.11). Adding all the three terms illustrated in item [(a)], we obtain the final expression for the induced localized current operator within a triangular plaquette [see Fig. 1(b) in the main text] as

$$\tilde{\mathcal{I}}_{ij,k}^{(2)} = \mathcal{I}_0(3S_i^x S_j^y + S_i^y S_j^x)S_k^z + \mathcal{I}_0(3S_i^y S_j^z - 5S_i^z S_j^y)S_k^x + \mathcal{I}_0(3S_i^z S_j^x - 5S_i^x S_j^z)S_k^y, \quad \text{(B.12a)}$$

$$\mathcal{I}_0 = \frac{e\hat{\mathbf{r}}_{ij}}{\hbar} \frac{8t_2^2 t_2' J_{\text{H}}(U - 2J_{\text{H}})}{9(U^2 - 4UJ_{\text{H}} + 3J_{\text{H}}^2)^2}. \quad \text{(B.12b)}$$

First, we notice that SU(2) symmetry of the spins is absent with the additional three coefficients $1, 3, 5$ and differ from the well-known form for the current operator in the SU(2) symmetric single-orbital Hubbard model [*see Eq. (1) in the main text*] [37]. Furthermore, the most important property of this structure is that it does not contain any repeated terms such as $S_i^\alpha S_j^\beta S_k^\gamma$, where $\{\alpha, \beta, \gamma\} = \{x, y, z\}$ can be equal to each other. It is an artifact of retaining only $t_2$ and $t_2'$ hopping terms in the Hamiltonian, which preserves the integrability of the Kitaev Hamiltonian. Inclusion of other hopping parameters like $t_1$, $t_3$ for more realistic modeling would modify this structure with additional three-spin terms with repeated spin indices [18, 24]. However, here we skip the analysis of the effect of the non-integrable terms in the Kitaev model and leave it for another future work.

Plugging in the parameter values from Sec. (B.3), we estimate the overall magnitude of the loop current as $|\mathcal{I}_0| \sim 30$ nA. Since this current flows around the sides of a triangle in a honeycomb plaquette, an induced magnetic field would appear in the center of the triangle, as shown in Fig. 4. Considering the lattice constant $a_0 = 3.44$ Å for $\alpha$-RuCl$_3$ [44] and assuming a finite expectation value of the induced circulating current operator in the ground state, we obtain an induced out-of-plane magnetic field at the center of the triangle as

$$\mathbf{B}_\perp = \frac{\mu_0}{4\pi} \int_C \frac{|\tilde{\mathcal{I}}_{ij,k}^{(2)}| d\mathbf{l} \times \hat{\mathbf{r}}}{\mathbf{r}^2}, \quad \text{(B.13)}$$

where C is the contour of the triangular plaquette. We consider $\mu_0 = 4\pi \times 10^{-7}$ NA$^{-2}$ as the free space magnetic permeability, and $\mathbf{r}$ is the distance from the sides of the triangle. After a straight-forward trigonometric and algebraic analysis, we evaluate $\mathbf{B}_\perp$ as

$$\boldsymbol{\mathcal{B}}_\perp = \frac{\mu}{2\pi} |\tilde{\mathcal{I}}_{ij,k}^{(2)}| \left( \frac{\sin \frac{\phi_1}{2}}{d_1} + \frac{2\sin 2\phi_2 + 2\sin \frac{\phi_2}{2}}{d_2} \right) \hat{\boldsymbol{c}}, \quad \text{(B.14)}$$

where $d_1 = \frac{a_0}{6} \tan \frac{\phi_2}{2}$, $d_2 = \frac{a_0}{2\sqrt{3}}$, $\phi_1 = 150°$, $\phi_2 = 30°$ in Fig. 4, and $\hat{\boldsymbol{c}}$ is the unit vector along the crystalline $c$-axis. Note that an equilateral triangular loop (with sides $2L$), carrying current of amplitude $\mathcal{I}$, would produce a similar magnetic field at its center as $\mathbf{B}_\perp = 9\mu_0\mathcal{I}/(4\pi L)\hat{\boldsymbol{c}}$.

# C  Low-energy effective model

In this section, we briefly outline the derivation of the second-order effective Hamiltonian [*see Eq. (4) in the main text*] and discuss its exact solution in Majorana representation [12]. From

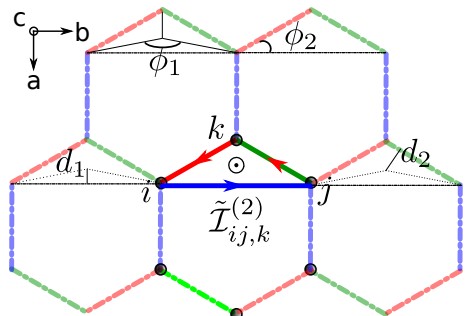

Figure 4: The triangular plaquette carrying the induced circulating current around its three sides (solid line). The induced magnetic field is out-of-plane (along the crystalline $c$-axis) as illustrated in the center of the triangle formed by the sites $i, j, k$. The respective angles $\phi_1$, and $\phi_2$ and the shortest distances from the center of the triangle to its sides $d_1$, and $d_2$ are illustrated, respectively.

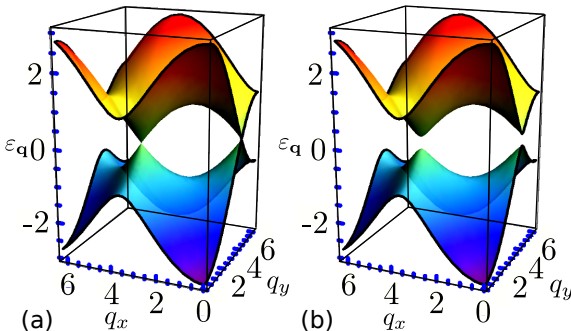

Figure 5: The gapless (a) and gapped (b) band structure of the real Majorana fermions obtained by diagonalizing the Hamiltonian in Eq. (C.5). Note that we illustrate the full Brillouin zone (BZ) although only half of the BZ contributes to the physical Hilbert space. The Dirac crossing point is illustrated by the band touching for the isotropic and homogeneous Kitaev model in panel (a). The parameter values for $K$ and $\kappa = 0.5K$ are inflated here in order to show the gap opening near the Dirac point in (b).

Eq. (A.6b) we obtain

$$\mathcal{H}_{\text{eff}}^{(2)} = \frac{i}{2}\mathcal{P}\mathcal{S}^{(1)}\mathcal{Q}_\Gamma\mathcal{Q}_\Gamma\mathcal{H}_1\mathcal{P} + \text{h.c.} = -\frac{1}{2}\sum_\Gamma\frac{1}{U_\Gamma}\mathcal{P}\mathcal{H}_1\mathcal{Q}_\Gamma\mathcal{Q}_\Gamma\mathcal{H}_1\mathcal{P} + \text{h.c.}, \tag{C.1}$$

where we utilized Eq. (A.8a) to recast the generating function $\mathcal{S}^{(1)}$ in terms of the hopping elements. Again using DiracQ package [82], we evaluate the matrix elements between the two sites $\langle ij \rangle$, and consequently obtain the effective Hamiltonian as

$$\mathcal{H}_0 = -K\sum_{\langle ij \rangle_{\mathbf{x}}} S_i^x S_j^x - K\sum_{\langle ij \rangle_{\mathbf{y}}} S_i^y S_j^y - K\sum_{\langle ij \rangle_{\mathbf{z}}} S_i^z S_j^z. \tag{C.2}$$

In the presence of an external magnetic field $\mathbf{h} = (h_x, h_y, h_z)$, a Zeeman term is introduced to Eq. (C.2) as $\mathcal{H}_{\text{mag}} = \sum_i \mathbf{h} \cdot \mathbf{S}_i$. However, in this case the total system $\mathcal{H}_0 + \mathcal{H}_{\text{mag}}$ becomes non-integrable. For a small magnetic field, we consider a low-energy effective form of $\mathcal{H}_{\text{mag}}$ [12], in perturbative expansion, as

$$\mathcal{H}_{\text{eff}} = \kappa \sum_{\substack{\langle ijk \rangle \\ \triangle}} S_i^x S_j^y S_k^z, \tag{C.3}$$

where $\kappa = h_x h_y h_z / K^2$, and $\Delta$ corresponds to sites within the enclosed triangle with $\langle ijk \rangle$. Utilizing Majorana representation $S_i^\alpha = i b_i^\alpha c_i / 2$, and link variables $u_{ij}$ as $u_{ij} = i b_i^\alpha b_j^\alpha$, Eq. (C.2) can be simplified as

$$\mathcal{H} = \mathcal{H}_0 + \mathcal{H}_{\text{eff}} = \frac{iK}{4} \sum_{\langle ij \rangle} u_{ij} c_i c_j + \frac{i\kappa}{8} \sum_{\langle\langle ij \rangle\rangle} u_{ik} u_{kj} c_i c_j = \frac{iK}{4} \sum_{\langle ij \rangle} c_i c_j + \frac{i\kappa}{8} \sum_{\langle\langle ij \rangle\rangle} c_i c_j \,, \tag{C.4}$$

where the link variables $u_{ij}$ have been set with their eigenvalues $+1$, as all $u_{ij}$'s commute with the Hamiltonian and hence can be considered conserved quantities. This particular choice of the link variables is equivalent to attaching zero-flux in each honeycomb plaquette [12]. Following Lieb's flux theorem [57], we identify this to be the ground state configuration for the underlying Majorana fermions [12]. Fourier transforming into the momentum-space, we obtain

$$\mathbf{H} = \sum_{\mathbf{q}} \mathbf{M}_{\mathbf{q}}^{A,B} c_{\mathbf{q},A} c_{-\mathbf{q},B} + \text{h.c.} \,, \quad \mathbf{M}_{\mathbf{q}} = \begin{pmatrix} \Delta_{\mathbf{q}} & i f_{\mathbf{q}} \\ -i f_{\mathbf{q}}^* & -\Delta_{\mathbf{q}} \end{pmatrix} \,, \tag{C.5}$$

where A, and B are the sub-lattice degrees of freedom. Two functions defined in the matrix $\mathcal{M}_{\mathbf{q}}$ are

$$f_{\mathbf{q}} = \frac{K}{4} \left( e^{i\mathbf{q} \cdot \mathbf{a}_1} + e^{i\mathbf{q} \cdot \mathbf{a}_2} + 1 \right) \,, \tag{C.6a}$$

$$\Delta_{\mathbf{q}} = \frac{\kappa}{4} \left[ \sin \mathbf{q} \cdot \mathbf{a}_1 - \sin \mathbf{q} \cdot \mathbf{a}_2 + \sin \mathbf{q} \cdot (\mathbf{a}_2 - \mathbf{a}_1) \right] \,, \tag{C.6b}$$

where $\mathbf{a}_i, i = 1, 2$ are the two basis vectors in the honeycomb lattice, given by $\mathbf{a}_1 = (1/2, \sqrt{3}/2)$ and $\mathbf{a}_2 = (-1/2, \sqrt{3}/2)$. Note that only half of the Brillouin zone (BZ) contributes to the physical Hilbert space as Majorana fermions are real quasiparticles with the property $c_{\mathbf{q},\alpha}^\dagger = c_{-\mathbf{q},\alpha}, \alpha = (A, B)$. The gapless (gapped) Dirac spectrum of the Majorana fermions is illustrated in Fig. 5, in the absence (presence) of an external magnetic field. In both the cases, the ground state in the extended Hilbert space is written as $|\Psi_0\rangle = |\mathcal{M}\rangle \otimes |\mathcal{G}_0\rangle$, where $|\mathcal{M}\rangle$ is the Majorana fermion ground state in a uniform gauge configuration $|\mathcal{G}_0\rangle$.

So far, we ignored the effect of the external magnetic field on the Hamiltonian Eq. (B.4c). However, in the presence of an external magnetic field, there will be the orbital coupling of the magnetic field term in the TB Hamiltonian through Peierl's substitution. Such a modification was not considered in the derivation of the induced loop current or the effective Kitaev Hamiltonian, as derived in Eq. (B.12a), and Eq. (C.2), respectively. If we consider such an orbital coupling of the external magnetic field, after a straightforward analysis as outlined in Sec. (B.3), we obtain a third-order contribution to the Hamiltonian in Eq. (C.2). The corresponding term can be written as

$$\mathcal{H}_{\text{eff}}^{(3)} = \sum_{\alpha,\beta,\gamma} \sin \left( \frac{\phi}{\phi_0} \right) \mathcal{A}_{\alpha,\beta,\gamma} S_i^\alpha S_j^\beta S_k^\gamma \,, \tag{C.7}$$

where the coefficients $\mathcal{A}_{\alpha,\beta,\gamma}$ can be obtained in a similar fashion as in Sec. (B.3), $\phi$ is the total flux within the triangular plaquette due to the external magnetic field, and $\phi_0 = \hbar c/e$ is the flux quantum [83]. For a small magnetic field ($\sim 10$ T) [32–34], this term is extremely small and we ignore it for the subsequent discussions.

Finally, we comment about the relation between the components of the external magnetic field $\mathbf{h} = (h_x, h_y, h_z)$ in the octahedral geometry and $\mathbf{h}^{\text{crys}} = h(\sin \theta \cos \phi, \sin \theta \sin \phi, \cos \theta)$ in the crystalline geometry. Doing a straightforward coordinate transformation between the

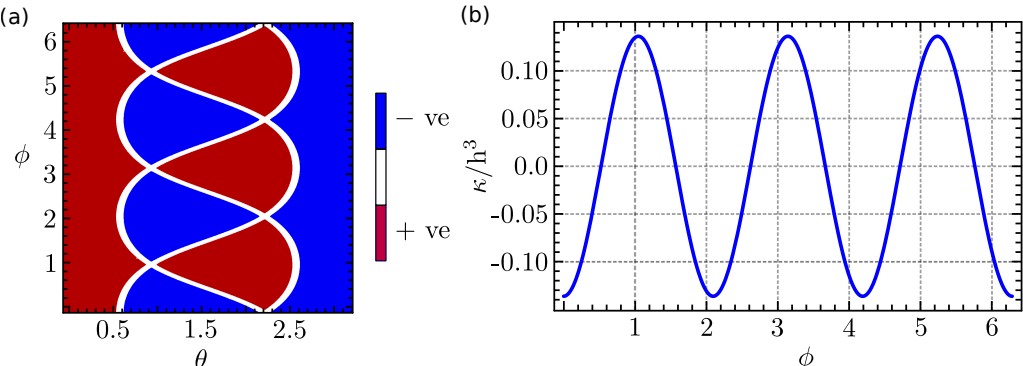

Figure 6: (a) Contour plot of $\text{sign}(\kappa)$ ($\kappa = h_x h_y h_z / K^2$) from Eq. (C.8) in the $\theta$-$\phi$ plane for the different orientations of the external magnetic field. (b) The variation of the magnitude of the Majorana gap function $\kappa$ as a function of the azimuthal angle measured from crystalline axis $a$ for an in-plane external magnetic field with $\theta = 90°$.

crystalline and octahedral geometry, we can write $\mathbf{h}$ [84]

$$
(h_x, h_y, h_z) = h\left( \frac{\cos\theta}{\sqrt{3}} + \frac{\sin\theta\cos\phi}{\sqrt{6}} - \frac{\sin\theta\sin\phi}{\sqrt{2}}, \right.
$$
$$
\left. \frac{\cos\theta}{\sqrt{3}} + \frac{\sin\theta\cos\phi}{\sqrt{6}} + \frac{\sin\theta\sin\phi}{\sqrt{2}}, \frac{\cos\theta}{\sqrt{3}} - \sqrt{\frac{2}{3}}\sin\theta\cos\phi \right), \qquad \text{(C.8)}
$$

where $\phi$ and $\theta$ are azimuthal angle measured from the crystalline $a$ axis and polar angle from the crystalline $c$ axis, respectively, and h is the strength of the applied magnetic field. Plugging it back in the Majorana gap $\kappa$ and considering an in-plane magnetic field ($\theta = 90°$), we obtain the variation of the strength of $\kappa$ upon a rotation of the in-plane magnetic field as illustrated in Fig. 6(a,b).

# D Computation: loop current expectation value

Now, we focus on the analysis for the evaluation of the expectation value of the loop current operator in the Kitaev ground state with Majorana representation. We first consider the uniform gauge configuration with all $u_{ij} = 1$. In this case, the analysis is simple with the Majorana fermion in the momentum-space representation. The target quantity is evaluating three-spin correlation functions *i.e.* $\langle S_i^\alpha S_j^\beta S_k^\gamma \rangle$, where $\{\alpha, \beta, \gamma\} = \{x, y, z\}$ with $\alpha \neq \beta \neq \gamma$ [*see Eq. (B.12b)*]. Along with the link variables, we further define the gauge invariant loop operators $\mathcal{W}_p = \prod_{\langle ij \rangle \in \bigcirc_p} u_{ij}$ in each hexagonal plaquette. In the uniform gauge, as mentioned earlier, $\mathcal{W}_p = 1$ for all the honeycomb cells. However, flipping one link variable in a single hexagon leads to $\mathcal{W}_p = -1$ in the two neighboring hexagons. The energy of the Majorana fermions increases with such non-uniform gauge configurations, and leads to the emergence of $\mathbb{Z}_2$ vortices, aka. visons [1, 2]. Each vison excitation creates a $\pi$-flux in the associated hexagon, and is static due to the integrable structure of the Kitaev model.

Hence, the non-zero expectation value of an arbitrary operator $\mathcal{O}_i$ exists only if the operator $\mathcal{O}_i$ conserves the vison occupation number. In this context, the action of three spin-operator $S_i^\alpha S_j^\beta S_k^\gamma$ on a ground state with finite/zero number vison excitations, has to preserve the vison occupation number. We note that a single spin operator $S_i^\alpha$ leads to two vison excitations and can be symbolically written as [85]

$$
S_i^\alpha \rightarrow i c_i \hat{\pi}_{1\langle ij \rangle_\alpha} \hat{\pi}_{2\langle ij \rangle_\alpha}, \qquad \text{(D.1)}
$$

where $\hat{\pi}_{1\langle ij\rangle_\alpha}$ and $\hat{\pi}_{2\langle ij\rangle_\alpha}$ are operators that introduce $\pi$-fluxes to the plaquettes 1 and 2 shared by the bond $\langle ij\rangle_\alpha$. Therefore, if the ground state of the Kitaev system does not have any vison excitations, a necessary condition for a finite expectation value of $S_i^\alpha S_j^\beta S_k^\gamma$ in this ground state, is that the two visons created by $S_k^\gamma$ should be destroyed by the other two spin operators.

## D.1 Pure Kitaev model in uniform gauge

In this case, the ground state $|\Psi_0\rangle$ was obtained from Eq. (C.5) where $|\Psi_0\rangle = |\mathcal{M}\rangle \otimes |\mathcal{G}_0\rangle$. Here, $|\mathcal{G}_0\rangle$ corresponds to the uniform gauge configuration without any vison excitations. After a straightforward algebra with Eq. (D.1), we see that only the first term in Eq. (A.10a) satisfy the constraint as mentioned in Appendix D. Hence, we have [using the Majorana representation of the spin operators]

$$\langle\Psi_0|\tilde{\mathcal{I}}_{ij,k}^{(2)}|\Psi_0\rangle = i\frac{\mathcal{I}_0}{8}\langle\Psi_0|c_i c_j|\Psi_0\rangle = i\frac{\mathcal{I}_0}{8}\langle\mathcal{M}|c_i c_j|\mathcal{M}\rangle, \tag{D.2}$$

where $|\mathcal{M}\rangle$ is the ground state of Majorana fermions. Choosing a convention that $i, j$ lie in sub-lattice B, we can rewrite $c_i c_j$ in momentum space as

$$
\begin{aligned}
c_{i,B} c_{j,B} &= \sum_{\mathbf{k},\mathbf{k}'} e^{i\mathbf{k}\cdot\mathbf{r}_i} e^{i\mathbf{k}'\cdot\mathbf{r}_j} c_{\mathbf{k},B} c_{\mathbf{k}',B} \\
&= \sum_{\substack{\mathbf{k},\mathbf{k}' \\ \in \\ \text{HBZ}}} e^{i\mathbf{k}\cdot\mathbf{r}_i} e^{i\mathbf{k}'\cdot\mathbf{r}_j} c_{\mathbf{k},B} c_{\mathbf{k}',B} + \sum_{\substack{\mathbf{k},\mathbf{k}' \\ \in \\ \text{HBZ}}} e^{-i\mathbf{k}\cdot\mathbf{r}_i} e^{-i\mathbf{k}'\cdot\mathbf{r}_j} c_{\mathbf{k},B}^\dagger c_{\mathbf{k}',B}^\dagger \\
&\quad + \sum_{\substack{\mathbf{k},\mathbf{k}' \\ \in \\ \text{HBZ}}} e^{-i\mathbf{k}\cdot\mathbf{r}_i} e^{i\mathbf{k}'\cdot\mathbf{r}_j} c_{\mathbf{k},B}^\dagger c_{\mathbf{k}',B} + \sum_{\substack{\mathbf{k},\mathbf{k}' \\ \in \\ \text{HBZ}}} e^{i\mathbf{k}\cdot\mathbf{r}_i} e^{-i\mathbf{k}'\cdot\mathbf{r}_j} c_{\mathbf{k},B} c_{\mathbf{k}',B}^\dagger,
\end{aligned}
\tag{D.3}
$$

where we defined the momentum summation in the first line over the full Brillouin zone (BZ) and reduced it to the half-Brillouin zone (HBZ) in the second line. Next, we represent the sub-lattice operators on the diagonal basis within the HBZ as

$$\begin{pmatrix} c_{\mathbf{k},A}^\dagger \\ c_{\mathbf{k},B}^\dagger \end{pmatrix} = \frac{1}{\sqrt{2}} \begin{pmatrix} -m_{\mathbf{k}} & m_{\mathbf{k}} \\ 1 & 1 \end{pmatrix} \begin{pmatrix} \alpha_{\mathbf{k}}^\dagger \\ \beta_{\mathbf{k}}^\dagger \end{pmatrix}, \tag{D.4}$$

where $m_{\mathbf{k}} = if_{\mathbf{k}}/|f_{\mathbf{k}}|$ from Eq. (C.5) (in the *absence* of any external magnetic field). Finally, the ground state is obtained by filling all the negative energy states and is written as

$$|\mathbf{M}\rangle = \prod_{\substack{\mathbf{k} \\ \in \\ \text{HBZ}}} \beta_{\mathbf{k}}^\dagger |0\rangle. \tag{D.5}$$

Consequently, a simple numerical integration (in Mathematica) using Eqs. (D.3-D.5) yields

$$\langle\Psi_0|\tilde{\mathcal{I}}_{ij,k}^{(2)}|\Psi_0\rangle = 0. \tag{D.6}$$

Hence, in the absence of any external magnetic field, the ground state of the pure Kitaev model does not allow any localized current expectation value. This result is consistent with the global time-reversal symmetry of the underlying system. Therefore, to induce a non-zero expectation value for the localized current operator, we break the time-reversal symmetry by applying an external magnetic field.

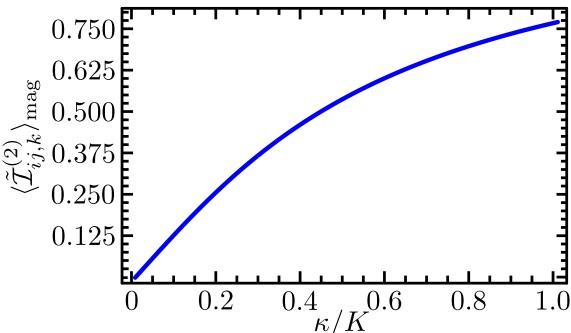

Figure 7: The variation of average localized current (in unit of $|\mathcal{I}_0|$) as a function of $\kappa$. For small $\kappa$ we notice the linear dependence of the current operator. *Note*: We utilized inflated values for $\kappa$ to illustrate the dependence.

### D.2 Kitaev model in external magnetic field

In the presence of a magnetic field, the Majorana fermions acquire a gap near the Dirac point [*see Fig. 5(b)*]. In this case, the ground state is written as $|\Psi_{\mathrm{mag}}\rangle = |\mathcal{M}_{\mathrm{mag}}\rangle \otimes |\mathcal{G}_0\rangle$, where $|\mathcal{M}_{\mathrm{mag}}\rangle$ is ground state of the Hamiltonian in Eq. (C.5). Writing the sub-lattice operators in terms of the diagonal operators we have

$$
\begin{pmatrix} c_{\mathbf{k},A}^{\dagger} \\ c_{\mathbf{k},B}^{\dagger} \end{pmatrix} = \begin{pmatrix} -i\frac{f_{\mathbf{k}}(\varepsilon_{\mathbf{k}}+\Delta_{\mathbf{k}})}{|f_{\mathbf{k}}|\sqrt{|f_{\mathbf{k}}|^2+(\varepsilon_{\mathbf{k}}+\Delta_{\mathbf{k}})^2}} & i\frac{f_{\mathbf{k}}(\varepsilon_{\mathbf{k}}-\Delta_{\mathbf{k}})}{|f_{\mathbf{k}}|\sqrt{|f_{\mathbf{k}}|^2+(\varepsilon_{\mathbf{k}}-\Delta_{\mathbf{k}})^2}} \\ \frac{|f_{\mathbf{k}}|}{\sqrt{|f_{\mathbf{k}}|^2+(\varepsilon_{\mathbf{k}}+\Delta_{\mathbf{k}})^2}} & \frac{|f_{\mathbf{k}}|}{\sqrt{|f_{\mathbf{k}}|^2+(\varepsilon_{\mathbf{k}}-\Delta_{\mathbf{k}})^2}} \end{pmatrix} \begin{pmatrix} \alpha_{\mathbf{k}}^{\dagger} \\ \beta_{\mathbf{k}}^{\dagger} \end{pmatrix}, \tag{D.7}
$$

where $\varepsilon_{\mathbf{k}} = \sqrt{|f_{\mathbf{k}}|^2 + \Delta_{\mathbf{k}}^2}$. In a similar fashion to Eq. (D.6), the Majorana fermion ground state is written in terms of the diagonal operators as

$$
|\mathbf{M}_{\mathrm{mag}}\rangle = \prod_{\substack{\mathbf{k} \\ \in \\ \mathrm{HBZ}}} \beta_{\mathbf{k}}^{\dagger}|0\rangle . \tag{D.8}
$$

Since the gauge configuration characterized by $|\mathcal{G}_0\rangle$ constraints all the gauge invariant Wilson loops $\mathcal{W}_p = 1$, only one of the six spin combination, in Eq. (B.12a), gives non-zero expectation value in the ground state $|\Psi_{\mathrm{mag}}\rangle$. Consequently, we have

$$
\langle \Psi_{\mathrm{mag}}|\tilde{\mathcal{I}}_{ij,k}^{(2)}|\Psi_{\mathrm{mag}}\rangle = i\frac{\mathcal{I}_0}{8} \langle \mathcal{M}_{\mathrm{mag}}|c_i c_j|\mathcal{M}_{\mathrm{mag}}\rangle . \tag{D.9}
$$

Expanding the Majorana bilinear operator $c_i c_j$ with Eq. (D.7) in the diagonal basis and after a straight-forward algebra, we obtain

$$
\langle \Psi_{\mathrm{mag}}|\tilde{\mathcal{I}}_{ij,k}^{(2)}|\Psi_{\mathrm{mag}}\rangle = i\frac{\mathcal{I}_0}{8}\mathrm{Re}\left( \sum_{\substack{\mathbf{k} \\ \in \\ \mathrm{HBZ}}} e^{i\mathbf{k}.(\mathbf{r}_i-\mathbf{r}_j)}\left[a_{\mathbf{k}}^2 + b_{\mathbf{k}}^2\right] - 2i b_{\mathbf{k}}^2 \sin\left[\mathbf{k}.(\mathbf{r}_i - \mathbf{r}_j)\right] \right), \tag{D.10}
$$

where $a_{\mathbf{k}} = \frac{|f_{\mathbf{k}}|}{\sqrt{|f_{\mathbf{k}}|^2+(\varepsilon_{\mathbf{k}}+\Delta_{\mathbf{k}})^2}}$, and $b_{\mathbf{k}} = \frac{|f_{\mathbf{k}}|}{\sqrt{|f_{\mathbf{k}}|^2+(\varepsilon_{\mathbf{k}}-\Delta_{\mathbf{k}})^2}}$. We perform the momentum integration in HBZ numerically in Mathematica. The corresponding variation of $\langle \tilde{\mathcal{I}}_{ij,k}^{(2)}\rangle_{\mathrm{mag}}$ as a function of the Majorana gap parameter $\kappa$ is shown in Fig. 6 (in unit of $|\mathcal{I}_0|$). For small $\kappa$ the linear dependence of the current operator becomes apparent.

# E Kitaev model in a finite system: Vison excitations and external magnetic field

In the previous two sections, we analyzed the expectation value of the current operator in the absence of any vison excitations in an infinite (periodic boundary condition) system. As mentioned earlier, this corresponds to choosing all the link variables $u_{ij} = 1$, with the constraint that $i$ in each bond $\langle ij \rangle$ belongs to the sub-lattice A. Flipping a particular bond $u_{ij}$ to $-1$ would, therefore, create two $\pi$-fluxes in the adjacent honeycomb plaquettes as explained in Appendix (D). We can separate these two adjacent $\mathbb{Z}_2$ vortices or visons by a string operator as illustrated in Fig. 8(a), to minimize their mutual interactions. Note that all the link variables defined on the bonds that cross the string operator are flipped. However, such vison configurations destroy the translational invariance of the Majorana system, and we cannot simply go to the momentum space to do our analysis.

Consequently, we first consider a spin-system on a 2D honeycomb lattice with $L \times L$ unit cells as shown in Fig. 8(a). Since each unit cell contain two sub-lattices (A & B), there are $2L^2$ Majorana operators $c_i$ in the system. The lattice vectors are chosen as $\mathbf{a}_1 = (1/2, \sqrt{3}/2)$, and $\mathbf{a}_2 = (-1/2, \sqrt{3}/2)$. The Majorana operators at a site $i$ is written as $c_i = c_\eta(m, n)$, where $\eta$ corresponds to the sub-lattice index, and $\mathbf{R}(m, n) = m\mathbf{a}_1 + n\mathbf{a}_2$, $m, n = 1, 2, \dots L$. Following Ref. [ [12]], we impose periodic boundary condition (PBC) as $c_\eta(m + L, n) = c_\eta(m, n)$, and $c_\eta(m, n + L) = c_\eta(m, n)$. The $2L^2$-dimensional Majorana vector is constructed as $\tilde{\mathbf{c}} = (c_A, c_B)^\mathsf{T}$ with

$$
\begin{aligned}
c_\eta = (&c_\eta(1, 1), c_\eta(2, 1), \dots c_\eta(L, 1), \\
&c_\eta(1, 2), c_\eta(2, 2), \dots c_\eta(L, 2), \dots, c_\eta(L, L))^\mathsf{T}, \quad \eta \in \{A, B\}.
\end{aligned} \tag{E.1}
$$

In terms of the Majorana vector $\tilde{\mathbf{c}}$, we can write the Kitaev model (in presence of an external magnetic field) as $\tilde{\mathbf{c}}^\mathsf{T} H \tilde{\mathbf{c}}$, where $H$ is written as

$$
\begin{aligned}
H = &i\frac{K}{4} \sum_{m,n} c_A(m, n) \Big( u_z(m, n)c_B(m, n) + u_x(m, n)c_B(m + 1, n) + u_y(m, n)c_B(m, n + 1) \Big) \\
&+ i\frac{\kappa}{8} \sum_{m,n} \Big( c_A(m, n) \big[ u_x(m, n)u_y(m + 1, n - 1)c_A(m + 1, n - 1) \\
&\qquad + u_z(m, n)u_x(m - 1, n)c_A(m - 1, n)u_y(m, n)u_z(m, n + 1)c_A(m, n + 1) \big] \\
&\quad + c_B(m, n) \big[ u_x(m - 1, n)u_y(m - 1, n)c_B(m - 1, n + 1) \\
&\qquad + u_z(m, n)u_x(m, n)c_B(m + 1, n) + u_y(m, n - 1)u_z(m, n - 1)c_B(m, n - 1) \big] \Big) \\
&+ \text{H.c.}, \tag{E.2}
\end{aligned}
$$

where $u_\alpha(m, n) = u_{\langle ij \rangle_\alpha}$, with $i$ ($j$) being in the sub-lattice A (B) as shown in Fig. 7(a). We now analyze the Majorana physics in a finite system with both the PBC and the open boundary condition (OBC), by diagonalizing the Hamiltonian in Eq. (E.2) in the real space. All numerical estimates will be written in the unit of Kitaev coupling $K$ in the subsequent sections.

## E.1 Kitaev model: Numerical convergence

At first, we diagonalize the above Hamiltonian in the *absence* of $\kappa$ and any vison excitations and analyze the ground state as a function of the system size $L$. The eigenmodes of the Hamiltonian in Eq. (E.2) can be obtained by a canonical transformation to a new set of Majorana

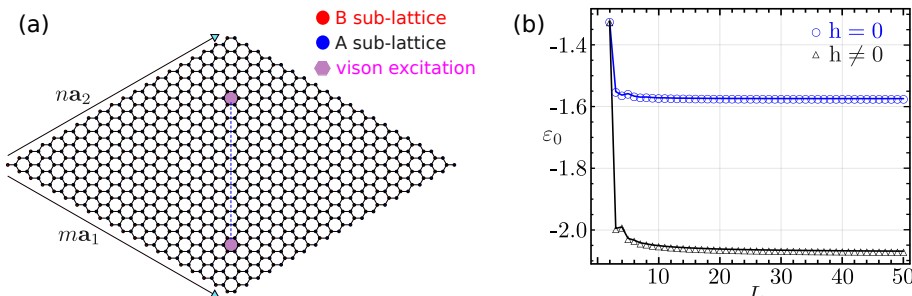

Figure 8: (a) A schematic of the finite size ($L \times L$, $L = 20$) honeycomb lattice. Two static vison excitations are located at the farthest distance $\lfloor \frac{L-1}{2} \rfloor$ in the middle of the system with periodic boundary condition (PBC). The link variables $u_{\langle ij \rangle}$ of the bonds crossing the blue dashed line (*string operator*) are all flipped to −1. (b) *No vison excitations:* Variation of the energy per unit cell $\varepsilon_0$ [*circles*: absence of external magnetic-field; *triangles*: non-vanishing external magnetic-field] as a function of the linear system size $L$ with PBC. Convergence is obtained for $L \sim 50$ ($L \sim 20$) in the absence (presence) of the external magnetic field. The energy is measured in the unit of $4K$. An inflated value of $\kappa = K$ is chosen for illustrative purposes.

operators [12] as

$$(b_1^{'}, b_1^{''}, \ldots b_N^{'}, b_N^{''}) = {}^{\mathsf{T}}\tilde{\mathcal{R}}, \tag{E.3}$$

where $\tilde{\mathcal{R}}$ is the canonical matrix. In terms of these operators the Hamiltonian in Eq. (E.1) can be rewritten as $\mathsf{H} = \frac{i}{2} \sum_m \varepsilon_m b_m^{'} b_m^{''}$, where $\varepsilon_m$ are the positive eigenvalues of H. Introducing fermions operators as $a_m = \frac{1}{2}(b_m^{'} + i b_m^{''})$, we have $\mathsf{H} = \sum_m \varepsilon_m (a_m^{\dagger} a_m - 1/2)$. Hence, the ground state energy is given by $\mathcal{E}_0 = -\frac{1}{2} \sum_m \varepsilon_m$. We now analyze the variation of average ground state energy $\varepsilon_0$ [$= 2\mathcal{E}_0/N^2$] per unit cell with the system size. The latter is shown in Fig. 8(b) [*empty circles*]. We notice that the average energy converges to 1.5746 (as noted by Kitaev himself [12]) for a linear system size of around $L \sim 50$. Performing a similar analysis in the presence of an external magnetic field, we notice that the convergence is achieved for a much smaller system with linear size $L \sim 20$. The variation of the energy per unit cell, in the external magnetic field, is illustrated by the (*empty triangles*) in Fig. 8(b).

Corresponding results for the convergence with system size in a periodic system and energy distribution in a finite system with PBC are shown in Fig. 7(b) and Fig. 7(c), respectively. Note that an inflated value of the magnetic field term $\kappa$ has been used in the numerical computation. However, the convergence is obtained at a much smaller system size with linear size $L \sim 20$, because of the gapped Majorana spectrum in Fig. 5(b).

Next, we introduce two $\pi$-fluxes in the adjacent honeycomb plaquettes shared by a bond $\langle ij \rangle$. In a periodic system, vison excitations can be only created in pairs since the string operator has to end inside the system as illustrated in Fig. 8(a). For this purpose, at first, we consider two adjacent plaquettes with $\pi$-fluxes which is achieved by putting the string operator on only one bond $\langle ij \rangle$. The energy of a single vison for two adjacent $\pi$-flux configuration comes out to be around $\sim 0.1311$. We compute the energy of a single vison by numerically computing the eigen-energy of the Hamiltonian with two visons and measuring the difference with the vison-free ground state as $(\mathcal{E}_2 - \mathcal{E}_0)/2$. Note that our quantitative estimate for the energy of a single vison is slightly less than Kitaev's original estimate ($\sim 0.1536$) [12], but is consistent with Ref. [59]. The analysis has been performed for a system of linear size $L = 50$ with two visons placed at the center of the system.

## E.2 Kitaev model in magnetic field: Current expectation value

Now we provide the main part of our analysis. We consider the Kitaev model in the presence of an external magnetic field which is parametrized by $\kappa$ in Eq. (E.2) for a finite system with both PBC and OBC. We consider two scenarios as mentioned in the main text: (a) the distribution of localized current in the system in absence of vison excitations (gauge sector $|\mathcal{G}_0\rangle$) in a finite system with OBC, and (b) the profile of localized current distribution around two vison excitations placed far away from the each other (gauge sector $|\mathcal{G}_2\rangle$) with PBC. For subsequent analysis it will be easier to transform the Majorana operators in Eq. (E.1) in terms of the matter fermions [85] as $f_i = c_{A,i} - ic_{B,i}$. On the latter basis, the $2L^2$-Majorana vector is transformed as

$$\tilde{c} = \begin{pmatrix} c_A & c_B \end{pmatrix}^\mathsf{T} = \mathsf{R} \begin{pmatrix} f & f^\dagger \end{pmatrix}^\mathsf{T}, \quad \mathsf{R} = \begin{pmatrix} \mathbb{I}_{L^2} & \mathbb{I}_{L^2} \\ i\mathbb{I}_{L^2} & -i\mathbb{I}_{L^2} \end{pmatrix}, \tag{E.4}$$

where $\mathbb{I}_{L^2}$ is an identity matrix of dimension $L^2$, and $f$-vector is defined as before

$$f = (f(1,1), f(2,1), \dots f(L,1), f(1,2), f(2,2), \dots f(L,2), \dots, f(L,L))^\mathsf{T}. \tag{E.5}$$

In terms of the matter fermion operators, the Hamiltonian in Eq. (E.2) can be rewritten as

$$\mathsf{H} = \begin{pmatrix} c_A & c_B \end{pmatrix} \begin{pmatrix} \mathcal{H}_{AA} & \mathcal{H}_{AB} \\ \mathcal{H}_{BA} & \mathcal{H}_{BB} \end{pmatrix} \begin{pmatrix} c_A \\ c_B \end{pmatrix} = \begin{pmatrix} f^\dagger & f \end{pmatrix} \mathsf{R}^\dagger \begin{pmatrix} \mathcal{H}_{AA} & \mathcal{H}_{AB} \\ \mathcal{H}_{BA} & \mathcal{H}_{BB} \end{pmatrix} \mathsf{R} \begin{pmatrix} f \\ f^\dagger \end{pmatrix}. \tag{E.6}$$

The block Hamiltonian $\mathcal{H}_{\alpha\beta}$ ($\alpha, \beta$ = A, B) is read off from Eq. (E.2). Instead of the canonical diagonalization as earlier, we now diagonalize the Hamiltonian in Eq. (E.6) by introducing the normal mode operators $a$, $a^\dagger$ as

$$\begin{pmatrix} f \\ f^\dagger \end{pmatrix} = U \begin{pmatrix} a \\ a^\dagger \end{pmatrix}, \tag{E.7}$$

where $U$ is a unitary matrix that diagonalizes the Hamiltonian H in Eq. (E.6). The resultant Hamiltonian in the $a$-basis reads as

$$\mathsf{H} = \begin{pmatrix} f^\dagger & f \end{pmatrix} \mathsf{R}^\dagger \begin{pmatrix} \mathcal{H}_{AA} & \mathcal{H}_{AB} \\ \mathcal{H}_{BA} & \mathcal{H}_{BB} \end{pmatrix} \mathsf{R} \begin{pmatrix} f \\ f^\dagger \end{pmatrix} = \begin{pmatrix} a^\dagger & a \end{pmatrix} \underbrace{U^\dagger \mathsf{R}^\dagger \begin{pmatrix} \mathcal{H}_{AA} & \mathcal{H}_{AB} \\ \mathcal{H}_{BA} & \mathcal{H}_{BB} \end{pmatrix} \mathsf{R} U}_{V_D} \begin{pmatrix} a \\ a^\dagger \end{pmatrix}, \tag{E.8}$$

where $V_D$ is the diagonal matrix with diagonal entries as the eigenenergies of the Hamiltonian H, and we obtain in terms of the $a$ fermions as

$$\mathsf{H} = \sum_{l=1}^{L^2} \varepsilon_l (2a_l^\dagger a_l - 1), \tag{E.9}$$

where $\varepsilon_l (\geq 0)$ are non-negative real eigenvalues of H ordered as $\varepsilon_1 < \varepsilon_2 < \cdots < \varepsilon_{L^2}$. Consequently, the ground state is defined by $|\mathcal{G}_0\rangle \otimes |\mathcal{M}_{mag}\rangle$ with the property $a_l |\mathcal{M}_{mag}\rangle = 0$. Here, gauge sector $|\mathcal{G}_0\rangle$ corresponds to an absence of any vison excitation in the system, and $|\mathcal{M}_{mag}\rangle$ is the Majorana fermion ground state in the presence of an external magnetic field. The ground state energy is given by $E_0 = -\sum_{l=1}^{L^2} \varepsilon_l$. We also define the bond fermion operator $\chi_{\langle ij \rangle_\alpha}$ as

$$\chi_{\langle ij \rangle_\alpha} = \frac{1}{2} \left( b_i^\alpha - ib_j^\alpha \right), \quad \chi_{\langle ij \rangle_\alpha}^\dagger = \frac{1}{2} \left( b_i^\alpha + ib_j^\alpha \right). \tag{E.10}$$

In terms of the bond fermions, the link variables can be recast as $u_{ij} = 1 - 2\chi_{\langle ij \rangle_\alpha}^\dagger \chi_{\langle ij \rangle_\alpha}$. Hence, for a uniform gauge configuration, with all $u_{ij} = 1$, we do not have any bond-fermions to start with. Flipping a specific link variable, hence, creates one bond fermion and associated

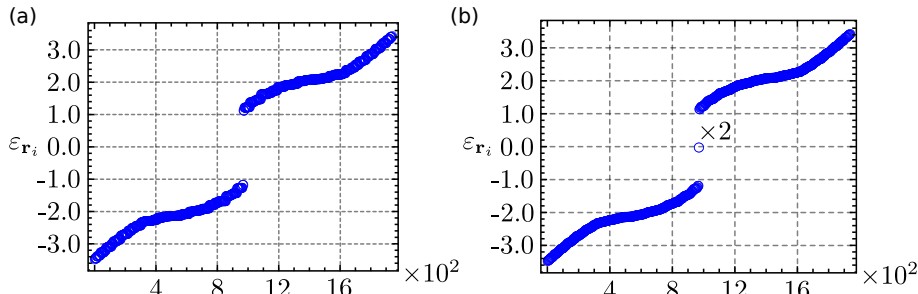

Figure 9: (a) The eigenvalue distribution as a function of the Hilbert space dimension for the Kitaev model in an external magnetic field in the uniform gauge configuration without any vison excitations. (b) The eigenvalue distribution in the same external magnetic field in the presence of two well-separated vison excitations with two Majorana zero modes (MZM) exactly at the zero energy.

two Majorana zero modes. The energy distribution for a finite system with PBC of linear size $L = 32$ is shown in Fig. 9(a) (9(b)) in the absence (presence) of vison excitations. For two vison excitations placed at a maximal distance, we have two Majorana zero modes (MZM) as illustrated in Fig. 9(b).

In terms of the new "$a$" fermions, we can now evaluate the expectation value of the loop current operator. The current operator can be written in a compact form as $\tilde{\mathcal{I}}^{(2)}_{ij,k} = \mathbf{A}_{\alpha\beta\gamma} S_i^\alpha S_j^\beta S_k^\gamma$, where the coefficients $\mathbf{A}_{\alpha\beta\gamma}$ are read off from Eq. (B.12a). With a particular choice of the gauge sector $|\mathcal{G}_0\rangle$ or $|\mathcal{G}_2\rangle$), we can recast the above current operator in terms of the Majorana fermions. Since the visons in the gauge sector are static, it leads to a huge simplification in computing the average of the current operator in the ground state. The non-zero expectation value of the terms in the ground state is only the terms in the current operator that preserve the number of the vison excitations. It turns out that only the first term in Eq. (B.12a) satisfies this condition.

On the other hand, since each bond in the honeycomb lattice is shared between four triangles, four terms will contribute to the total average current flowing in the bond. Consequently, we obtain the current expression for a particular bond $\alpha$ on the honeycomb plaquette as

$$\mathcal{I}_\alpha = \sum_\triangle \langle \Psi | \tilde{\mathcal{I}}^{(2)}_{ij,k} | \Psi \rangle = 3\mathcal{I}_0 \sum_{k,\triangle} u_{ik} u_{kj} \langle\langle c_i c_j \rangle\rangle - \text{"neighboring plaquette"}, \tag{E.11}$$

where the indices $ij, k$ in each triangle are arranged according to the orientation of the associated triangles, and $\langle\langle c_i c_j \rangle\rangle$ signifies that expectation value of the operator $c_i c_j$ when the sites $i, j$ are connected by the next nearest-neighbor bonds on the associated triangle. The "neighboring plaquette" term corresponds to the triangles belonging to the other honeycomb plaquette shared by the bond $\alpha$.

Hence, the analysis of the expectation of the current operator eventually boils down to the computation of the expectation value of the product of Majorana operators $\langle c_A(\mathbf{R}_{mn}) c_A(\mathbf{R}'_{m'n'}) \rangle$ and $\langle c_B(\mathbf{R}_{mn}) c_B(\mathbf{R}'_{m'n'}) \rangle$, where $\mathbf{r}_{mn}$ denotes the position of the site at $\mathbf{R}_{mn} = m\hat{\mathbf{a}}_1 + n\hat{\mathbf{a}}_2$. Considering a gauge configuration $|\mathcal{G}_0\rangle$ or $|\mathcal{G}_1\rangle$, such expectation can be computed as following

$$\langle c_\alpha(\mathbf{R}_{mn}) c_\alpha(\mathbf{R}'_{m'n'}) \rangle = \left\langle \begin{pmatrix} c_A & c_B \end{pmatrix} \mathcal{W}^{(\alpha)}_{\mathbf{R}\mathbf{R}'} \begin{pmatrix} c_A \\ c_B \end{pmatrix} \right\rangle = \sum_{ll'=1}^{2L^2} \left( U^\dagger R^\dagger \mathcal{W}^{(\alpha)}_{\mathbf{R}\mathbf{R}'} R U \right)_{ll'} \langle a_l^\dagger a_{l'} \rangle$$

$$= \sum_{l=L^2+1}^{2L^2} \left( U^\dagger R^\dagger \mathcal{W}^{(\alpha)}_{\mathbf{R}\mathbf{R}'} R U \right)_{ll}, \tag{E.12}$$

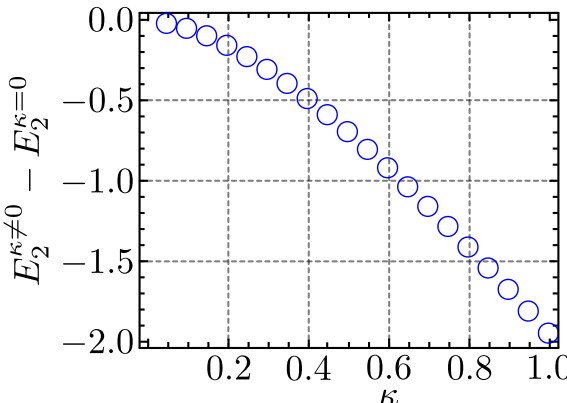

Figure 10: The difference between the Majorana fermion ground energy for two vison configuration [with well-separated visons as shown in Fig. 8(a)] in the presence and absence of an external magnetic field plotted as a function of $\kappa$. The ground state energy decreases with increasing magnetic field strength.

where $\mathcal{W}_{\mathbf{RR'}}^{(\alpha)}$ is $2L^2 \times 2L^2$ matrix defined as follows

$$\mathcal{W}_{\mathbf{RR'}}^{(\alpha)} = \begin{pmatrix} \delta_{\alpha,\mathrm{A}} \mathcal{B}_{\mathbf{RR'}}^{(\alpha)} & \mathcal{O}_{L^2} \\ \mathcal{O}_{L^2} & \delta_{\alpha,\mathrm{B}} \mathcal{B}_{\mathbf{RR'}}^{(\alpha)} \end{pmatrix}. \tag{E.13}$$

Here, $\mathcal{B}_{\mathbf{RR'}}^{(\alpha)}$ are the matrices corresponding to the non-zero connections allowed by the orientations of the triangles, and $\mathcal{O}_{L^2}$ is a null matrix of order $L^2 \times L^2$. We compute the above expectation value for both the gauge configurations $|\mathcal{G}_0\rangle$, and $|\mathcal{G}_2\rangle$. Depending on the number of the bond-fermions, we may have to consider the effects of non-trivial Majorana zero modes and the issue with the bond fermions. Following Ref. [59, 60], then we have to satisfy the parity constraint for the matter fermions as $(-1)^{n_f + n_\chi} = 1$, where $n_f$ ($n_\chi$) denotes the number of matter (bond) fermions. Depending on the situation, we may have to modify the above Eq. E.12 as follows

$$\langle c_\alpha(\mathcal{R}_{mn}) c_\alpha(\mathcal{R}'_{m'n'}) \rangle = \sum_{l=L^2+2}^{2L^2} \left( U^\dagger \mathsf{R}^\dagger \mathcal{W}_{\mathbf{RR'}}^{(\alpha)} \mathsf{R} U \right)_{ll} + \left( U^\dagger \mathsf{R}^\dagger \mathcal{W}_{\mathbf{RR'}}^{(\alpha)} \mathsf{R} U \right)_{L^2 L^2}, \tag{E.14}$$

to consider one matter fermion excitation. In the presence of MZM with two well-separated visons, we perform singular value decomposition (SVD) to diagonalize the Hamiltonian matrix in Eq. (E.6). Summing over the triangles as mentioned earlier, we obtain the localized current profiles as illustrated in Fig. 1 and Fig. 2, in the main text. In the thermodynamic limit, when the visons are far away from each other, the current profile becomes identical around each hexagonal plaquette containing the vison excitations. Hence, we focus our analysis around a single vison excitation.

Furthermore, we note that in two-vison configuration $|\mathcal{G}_2\rangle$, the ground state energy of the Majorana fermions decreases in the presence of an external magnetic field. In Fig. 10, we show the variation of ground state energy $E_2^{\kappa \neq 0} - E_2^{\kappa = 0}$ for various strength and orientation of the magnetic field. As the energy difference decreases in the presence of the magnetic field, the vison configuration is favored in the presence of the external magnetic field. Hence, the magnetic field lowers the energy for the vison configurations and may lead to formation of exotic vison crystal phases [86]. We leave such an explicit analysis for future work.

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
