# Peer review of "Emergent orbital magnetization in Kitaev quantum magnets"

_SciPost Physics, doi:SciPost Phys. 14, 127 (2023)_

## Round 1 · Referee Report · Anonymous (Referee 1) · 2022-10-6

Strengths

1- The paper is well written and clearly motivated. 2- The results combine a phenomenological picture with a detailed derivation for the microscopic model. 3- Nice interpretation of the numerical results in terms of symmetries.
4- The authors predict a new effect in Kitaev materials (orbital magnetic moment perpendicular to the applied field).

Weaknesses

1- Unfortunately the effect is rather small and unlikely to be relevant for experiments, at least for RuCl3.

Report

In this work the authors study orbital currents in Kitaev materials. They build on previous results by Bulaevskii et al. (Ref. [37]), who derived an effective orbital current operator for the Mott-insulating phase of the single-orbital Hubbard model using a Schrieffer-Wolff transformation. Here the authors go through the same kind of calculation for the Hubbard-Kanamori model that describes alpha-RuCl3 and other materials that may exhibit a Kitaev spin liquid phase at intermediate magnetic fields. Having derived the expression for the orbital current in terms of spin operators, the authors compute its expectation value in the ground state of the Kitaev model on a finite lattice with open boundary conditions, relying on the exact solution of the model in terms of Majorana fermions and a static Z2 gauge field. They discuss the orbital current pattern in the bulk as well as near the edge, and also in the vicinity of a Z2 vortex. Among their main results, they find that an in-plane magnetic field can induce an orbital magnetic moment perpendicular to the honeycomb plane, and propose this effect as a signature of the Kitaev spin liquid phase.

Nontrivial electrical responses of Kitaev spin liquids have been discussed in a few recent papers (Refs. [39,40]). To my knowledge the results for the orbital current presented here are new. The analytical expression for the orbital current operator and the numerical results are all consistent with symmetry considerations and the conclusions are quite interesting. While the effects of the orbital currents turn out to be rather small for realistic parameters, I believe this work presents an important contribution to the list of possible signatures of the Kitaev spin liquid phase.

In my opinion the paper meets the journal's publication criteria.

Requested changes

Before publication, I would like the authors to consider the following points:

  1. In the introduction, the authors mention that inelastic neutron scattering experiments have observed a finite spin gap at magnetic fields above 10 T. To avoid misunderstanding, it should be acknowledged that a spin gap is also expected for the partially polarized phase at high fields, thus the spin gap by itself is not evidence of a Kitaev spin liquid phase.

  2. Perturbation theory in the next-nearest-neighbor hopping t2' should generate a next-nearest-neighbor interaction beyond the exactly solvable model in Eq. (4). The authors should explain the conditions for neglecting this interaction.

  3. In the last paragraph of section III, the statement "The seemingly two-fold rotation ... is absent when one embeds the honeycomb plane into the parent crystal" is not clear to me. It would be helpful if the authors could elaborate on what specific property of the crystal breaks this symmetry, and how this property is manifested in the effective spin model.

  4. On page 7, the authors refer to Fig. 1(a) concerning the edge E1, but I believe they mean Fig. 2(a).

  5. Since the cancellation of orbital currents on nearest neighbour bonds in the bulk depends on translational invariance, in principle it could be violated by any kind of inhomogeneity, for instance bond disorder. Perhaps the authors could add a comment about how their conclusions might be modified in the presence of disorder.

  6. In section VI, the authors refer to the out-of-plane orbital moment as "ferromagnetic order within the Kitaev QSL phase". The term is confusing because it suggests spontaneous symmetry breaking, but if I understand correctly this is not the case because the nonzero moment is allowed once time reversal and lattice rotation symmetries have been broken.

  7. The conclusion that visons carry a physical magnetic flux is quite interesting. Is it possible to estimate the change in the orbital magnetic moment associated with a vison (with respect to the magnetic moment of a hexagon in the uniform ground state)?

  8. The arXiv version includes a supplementary material. I assume that this material will not be published with the main text, otherwise it should be substantially shortened and turned into appendices.

  • validity: high
  • significance: high
  • originality: good
  • clarity: high
  • formatting: excellent
  • grammar: excellent

Author:  Shizeng Lin  on 2023-01-05  [id 3209]

(in reply to Report 1 on 2022-10-06)

The Reviewer wrote: In this work the authors study orbital currents in Kitaev materials. They build on previous results by Bulaevskii et al. (Ref. [37]), who derived an effective orbital current operator for the Mott-insulating phase of the single-orbital Hubbard model using a Schrieffer-Wolff transformation. Here the authors go through the same kind of calculation for the Hubbard-Kanamori model that describes alpha-RuCl3 and other materials that may exhibit a Kitaev spin liquid phase at intermediate magnetic fields. Having derived the expression for the orbital current in terms of spin operators, the authors compute its expectation value in the ground state of the Kitaev model on a finite lattice with open boundary conditions, relying on the exact solution of the model in terms of Majorana fermions and a static Z2 gauge field. They discuss the orbital current pattern in the bulk as well as near the edge, and also in the vicinity of a Z2 vortex. Among their main results, they find that an in-plane magnetic field can induce an orbital magnetic moment perpendicular to the honeycomb plane, and propose this effect as a signature of the Kitaev spin liquid phase.
Nontrivial electrical responses of Kitaev spin liquids have been discussed in a few recent papers (Refs. [39,40]). To my knowledge the results for the orbital current presented here are new. The analytical expression for the orbital current operator and the numerical results are all consistent with symmetry considerations and the conclusions are quite interesting. While the effects of the orbital currents turn out to be rather small for realistic parameters, I believe this work presents an important contribution to the list of possible signatures of the Kitaev spin liquid phase.
In my opinion the paper meets the journal's publication criteria.
Our reply: We thank the Reviewer for appreciating our work and for their recommendation for publication in SciPost. Herein-below, we provide a detailed response to the comments and suggestions raised by the Reviewer.

Reviewer's comment (#1): In the introduction, the authors mention that inelastic neutron scattering experiments have observed a finite spin gap at magnetic fields above 10 T. To avoid misunderstanding, it should be acknowledged that a spin gap is also expected for the partially polarized phase at high fields, thus the spin gap by itself is not evidence of a Kitaev spin liquid phase.
Our reply: We agree with this statement, and have modified our manuscript in the introduction explaining the inconclusive nature of the observed spin gap in neutron scattering experiments regarding the determination of spin liquid phase.

Reviewer's comment (#2): Perturbation theory in the next-nearest-neighbor hopping t2' should generate a next-nearest-neighbor interaction beyond the exactly solvable model in Eq. (4). The authors should explain the conditions for neglecting this interaction.
Our reply: This is an important point. The referee is correct that the inclusion of next-nearest-neighbor hopping t2' would lead to further neighbor interaction in the Hamiltonian, through a similar second- order perturbation expansion. We treat t2' as a small perturbation in our analysis. Therefore, we first obtain the ground state configuration without t2'. Then the induced magnetization is obtained by considering t2' as a small perturbation in the expression for the orbital current in terms of spin operators. Our analysis is accurate up to the first order in t2'. We have added several sentences in the manuscript to clarify this point.

Reviewer's comment (#3): In the last paragraph of section III, the statement "The seemingly two-fold rotation ... is absent when one embeds the honeycomb plane into the parent crystal" is not clear to me. It would be helpful if the authors could elaborate on what specific property of the crystal breaks this symmetry, and how this property is manifested in the effective spin model.
Our reply: For clarity, we did not show the ligands in Fig. 1 (a). In real materials, the magnetic ions are caged by otahedral structure of ligands, which breaks the C2a symmetry. As an example, the crystal symmetry for the mono-layer α-RuCl3 is P63/mcm, which does not contain C2a symmetry. We added one sentence in the caption of Fig. 1 to clarify this point.

Reviewer's comment (#4): On page 7, the authors refer to Fig. 1(a) concerning the edge E1, but I believe they mean Fig. 2(a).
Our reply: Corrected. We apologize for this typo.

Reviewer's comment (#5): Since the cancellation of orbital currents on nearest neighbour bonds in the bulk depends on translational invariance, in principle it could be violated by any kind of inhomogeneity, for instance bond disorder. Perhaps the authors could add a comment about how their conclusions might be modified in the presence of disorder.
Our reply: We agree that a perfect cancellation of the localized orbital currents on the nearest neighbor (NN) bonds requires the translational invariance of the system, and local inhomogeneties (which breaks this symmetry) would lead to the modification of the current profile including appearance of currents on the NN bonds. The current pattern induced by impurities can be random and is different from the current pattern induced by a vison calculated in the manuscript. Furthermore, visons are dynamical excitations of the quantum spin liquid, while the current induced by impurities is static. These two distinct features for current induced by impurities and visons can be distinguished experimentally. We have added several sentences to the manuscript to explain this subtle difference.

Reviewer's comment (#6): In section VI, the authors refer to the out-of-plane orbital moment as "ferromagnetic order within the Kitaev QSL phase". The term is confusing because it suggests spontaneous symmetry breaking, but if I understand correctly this is not the case because the nonzero moment is allowed once time reversal and lattice rotation symmetries have been broken.
Our reply: We have rephrased this sentence to "induced out-of-plane (along c-axis) magnetization" to avoid potential confusion.

Reviewer's comment (#7): The conclusion that visons carry a physical magnetic flux is quite interesting. Is it possible to estimate the change in the orbital magnetic moment associated with a vison (with respect to the magnetic moment of a hexagon in the uniform ground state)?
Our reply: We thank the referee for this point. We have updated our manuscript by explicitly mentioning the relative change of the magnitude of the magnetization around a vison excitation compared to the uniform case.

Reviewer's comment (#8): The arXiv version includes a supplementary material. I assume that this material will not be published with the main text, otherwise it should be substantially shortened and turned into appendices.
Our reply: Supplementary materials are allowed according to some published SciPost papers. Therefore we keep the supplementary material as it is.

---

## Round 1 · Referee Report · Anonymous (Referee 2) · 2022-12-19

Report

The manuscript by Banerjee and Lin analyzes the orbital magnetization in Kitaev materials by carrying out a Schrieffer-Wolff transformation in an external magnetic field. There is a long history of pointing out such effects in spin-liquid-like materials in an external magnetic field, which have demonstrated that even electrically neutral quasiparticles/local-moments can develop orbital effects. The authors seem to have carried out an extension of this program to the present setting of Kitaev materials. I think the authors have included many well-known results in the introductory sections of the paper, which lends a pedagogical value to the manuscript. However, in that regard, I have two comments that the authors should address:

1) One of the surprising aspects of this work is the conspicuous omission (from the v1 version on arXiv) of an important earlier work by O. Motrunich, Phys. Rev. B 73, 155115, (2006). They have cited many subsequent works, which cite the above paper. The authors should cite this work and clarify the aspects of their paper that are clearly different (apart from the “Kitaev” aspect).

2) In the section on “phenomenological picture”, the authors have reproduced a rather standard argument (due originally to Ioffe-Larkin, which they do not cite), which others have used to evaluate the orbital effects (including e.g. the paper mentioned in point 1 above). However, it is unclear why the authors pick a relativistic theory for the b-boson in Eqn. (2). There is no a priori reason for this to be the case–after all not all Mott insulators necessarily have a relativistic gapped boson.

I have not tried to reproduce the detailed results obtained by the authors. They appear to be a natural extension of many previous works, and it seems the authors have done a careful job. While I don’t think the work is necessarily “novel” (by unfortunately a subjective metric), there are possibly readers who can benefit from the results in the manuscript. If the authors address all the concerns by the referees, it could eventually be accepted for publication.

  • validity: high
  • significance: good
  • originality: ok
  • clarity: high
  • formatting: excellent
  • grammar: excellent

Author:  Shizeng Lin  on 2023-01-05  [id 3208]

(in reply to Report 2 on 2022-12-19)

The Reviewer wrote: The manuscript by Banerjee and Lin analyzes the orbital magnetization in Kitaev materials by carrying out a Schrieffer-Wolff transformation in an external magnetic field. There is a long history of pointing out such effects in spin-liquid-like materials in an external magnetic field, which have demonstrated that even electrically neutral quasiparticles/local-moments can develop orbital effects. The authors seem to have carried out an extension of this program to the present setting of Kitaev materials. I think the authors have included many well-known results in the introductory sections of the paper, which lends a pedagogical value to the manuscript. However, in that regard, I have two comments that the authors should address:
Our reply: We thank the referee for their important comments and feedback on our work. Here-in- below, we address their comments in details.

Reviewer's comment (#1): One of the surprising aspects of this work is the conspicuous omission (from the v1 version on arXiv) of an important earlier work by O. Motrunich, Phys. Rev. B 73, 155115, (2006). They have cited many subsequent works, which cite the above paper. The authors should cite this work and clarify the aspects of their paper that are clearly different (apart from the “Kitaev” aspect).

Our reply: We have duly cited the work by O. Motrunich, Phys. Rev. B 73, 155115 (2006) in the modified version of our manuscript. This paper discusses the orbital response of a quantum spin liquid with spinon Fermi surface and the possibility of quantum oscillations of this state. While our work focuses on the orbital current in the Kiteav quantum spin liquid and the associated excitations. We feel that the work by O. Motrunich is not directly related to what we discussed in the manuscript.

Reviewer's comment (#2): In the section on “phenomenological picture”, the authors have reproduced a rather standard argument (due originally to Ioffe-Larkin, which they do not cite), which others have used to evaluate the orbital effects (including e.g. the paper mentioned in point 1 above). However, it is unclear why the authors pick a relativistic theory for the b-boson in Eqn. (2). There is no a priori reason for this to be the case–after all not all Mott insulators necessarily have a relativistic gapped boson.
Our reply: We are sorry for the confusion. We did not mean to include the temporal component of the gauge field because it corresponds to screening. The indices μ in Eq. (2) denote the spatial components. We have modified the manuscript to clarify this point. We also cited the paper by Ioffe-Larkin in the revised manuscript in Ref. [47].

Reviewer's wrote: I have not tried to reproduce the detailed results obtained by the authors. They appear to be a natural extension of many previous works, and it seems the authors have done a careful job. While I don’t think the work is necessarily “novel” (by unfortunately a subjective metric), there are possibly readers who can benefit from the results in the manuscript. If the authors address all the concerns by the referees, it could eventually be accepted for publication.
Our reply: We appreciate referee's decision about our work, and believe that the present version addresses all the concerns by the referees.

---

## Round 2 · Author Response

Dear Editor,
Thank you for forwarding us the referees’ reports. We thank the referees for their time and effort to review our manuscript entitled “Emergent orbital magnetization in Kitaev quantum magnets” and for their positive feedback.
Both referees gave positive assessments of our manuscript. At the same time, they raised several questions, either technical or clarification in nature. Below, we provide a detailed reply to the questions and comments raised by each of the referees and hereby resubmit a modified version of the manuscript with all the changes as requested by them.
We hope that the current version will be suitable for publication in Sci Posts Physics.
Yours sincerely,
Saikat Banerjee, and Shi-Zeng Lin
Los Alamos National Laboratory

---

## Round 2 · List of Changes

---------------Report of Referee #1 – 2208.06887v1/Banerjee et al. ---------------------------------------------
The Reviewer wrote: In this work the authors study orbital currents in Kitaev materials. They build on previous results by Bulaevskii et al. (Ref. [37]), who derived an effective orbital current operator for the Mott-insulating phase of the single-orbital Hubbard model using a Schrieffer-Wolff transformation. Here the authors go through the same kind of calculation for the Hubbard-Kanamori model that describes alpha-RuCl3 and other materials that may exhibit a Kitaev spin liquid phase at intermediate magnetic fields. Having derived the expression for the orbital current in terms of spin operators, the authors compute its expectation value in the ground state of the Kitaev model on a finite lattice with open boundary conditions, relying on the exact solution of the model in terms of Majorana fermions and a static Z2 gauge field. They discuss the orbital current pattern in the bulk as well as near the edge, and also in the vicinity of a Z2 vortex. Among their main results, they find that an in-plane magnetic field can induce an orbital magnetic moment perpendicular to the honeycomb plane, and propose this effect as a signature of the Kitaev spin liquid phase.
Nontrivial electrical responses of Kitaev spin liquids have been discussed in a few recent papers (Refs. [39,40]). To my knowledge the results for the orbital current presented here are new. The analytical expression for the orbital current operator and the numerical results are all consistent with symmetry considerations and the conclusions are quite interesting. While the effects of the orbital currents turn out to be rather small for realistic parameters, I believe this work presents an important contribution to the list of possible signatures of the Kitaev spin liquid phase.
In my opinion the paper meets the journal's publication criteria.
Our reply: We thank the Reviewer for appreciating our work and for their recommendation for publication in SciPost. Herein-below, we provide a detailed response to the comments and suggestions raised by the Reviewer.

Reviewer's comment (#1): In the introduction, the authors mention that inelastic neutron scattering experiments have observed a finite spin gap at magnetic fields above 10 T. To avoid misunderstanding, it should be acknowledged that a spin gap is also expected for the partially polarized phase at high fields, thus the spin gap by itself is not evidence of a Kitaev spin liquid phase.
Our reply: We agree with this statement, and have modified our manuscript in the introduction explaining the inconclusive nature of the observed spin gap in neutron scattering experiments regarding the determination of spin liquid phase.

Reviewer's comment (#2): Perturbation theory in the next-nearest-neighbor hopping t2' should generate a next-nearest-neighbor interaction beyond the exactly solvable model in Eq. (4). The authors should explain the conditions for neglecting this interaction.
Our reply: This is an important point. The referee is correct that the inclusion of next-nearest-neighbor hopping t2' would lead to further neighbor interaction in the Hamiltonian, through a similar second- order perturbation expansion. We treat t2' as a small perturbation in our analysis. Therefore, we first obtain the ground state configuration without t2'. Then the induced magnetization is obtained by considering t2' as a small perturbation in the expression for the orbital current in terms of spin operators. Our analysis is accurate up to the first order in t2'. We have added several sentences in the manuscript to clarify this point.

Reviewer's comment (#3): In the last paragraph of section III, the statement "The seemingly two-fold rotation ... is absent when one embeds the honeycomb plane into the parent crystal" is not clear to me. It would be helpful if the authors could elaborate on what specific property of the crystal breaks this symmetry, and how this property is manifested in the effective spin model.
Our reply: For clarity, we did not show the ligands in Fig. 1 (a). In real materials, the magnetic ions are caged by otahedral structure of ligands, which breaks the C2a symmetry. As an example, the crystal symmetry for the mono-layer α-RuCl3 is P63/mcm, which does not contain C2a symmetry. We added one sentence in the caption of Fig. 1 to clarify this point.

Reviewer's comment (#4): On page 7, the authors refer to Fig. 1(a) concerning the edge E1, but I believe they mean Fig. 2(a).
Our reply: Corrected. We apologize for this typo.

Reviewer's comment (#5): Since the cancellation of orbital currents on nearest neighbour bonds in the bulk depends on translational invariance, in principle it could be violated by any kind of inhomogeneity, for instance bond disorder. Perhaps the authors could add a comment about how their conclusions might be modified in the presence of disorder.
Our reply: We agree that a perfect cancellation of the localized orbital currents on the nearest neighbor (NN) bonds requires the translational invariance of the system, and local inhomogeneties (which breaks this symmetry) would lead to the modification of the current profile including appearance of currents on the NN bonds. The current pattern induced by impurities can be random and is different from the current pattern induced by a vison calculated in the manuscript. Furthermore, visons are dynamical excitations of the quantum spin liquid, while the current induced by impurities is static. These two distinct features for current induced by impurities and visons can be distinguished experimentally. We have added several sentences to the manuscript to explain this subtle difference.

Reviewer's comment (#6): In section VI, the authors refer to the out-of-plane orbital moment as "ferromagnetic order within the Kitaev QSL phase". The term is confusing because it suggests spontaneous symmetry breaking, but if I understand correctly this is not the case because the nonzero moment is allowed once time reversal and lattice rotation symmetries have been broken.
Our reply: We have rephrased this sentence to "induced out-of-plane (along c-axis) magnetization" to avoid potential confusion.

Reviewer's comment (#7): The conclusion that visons carry a physical magnetic flux is quite interesting. Is it possible to estimate the change in the orbital magnetic moment associated with a vison (with respect to the magnetic moment of a hexagon in the uniform ground state)?
Our reply: We thank the referee for this point. We have updated our manuscript by explicitly mentioning the relative change of the magnitude of the magnetization around a vison excitation compared to the uniform case.

Reviewer's comment (#8): The arXiv version includes a supplementary material. I assume that this material will not be published with the main text, otherwise it should be substantially shortened and turned into appendices.
Our reply: Supplementary materials are allowed according to some published SciPost papers. Therefore we keep the supplementary material as it is.

-------------------- Report of Referee #2 – 2208.06887v1/Banerjee et al.-----------------------------
The Reviewer wrote: The manuscript by Banerjee and Lin analyzes the orbital magnetization in Kitaev materials by carrying out a Schrieffer-Wolff transformation in an external magnetic field. There is a long history of pointing out such effects in spin-liquid-like materials in an external magnetic field, which have demonstrated that even electrically neutral quasiparticles/local-moments can develop orbital effects. The authors seem to have carried out an extension of this program to the present setting of Kitaev materials. I think the authors have included many well-known results in the introductory sections of the paper, which lends a pedagogical value to the manuscript. However, in that regard, I have two comments that the authors should address:
Our reply: We thank the referee for their important comments and feedback on our work. Here-in- below, we address their comments in details.

Reviewer's comment (#1): One of the surprising aspects of this work is the conspicuous omission (from the v1 version on arXiv) of an important earlier work by O. Motrunich, Phys. Rev. B 73, 155115, (2006). They have cited many subsequent works, which cite the above paper. The authors should cite this work and clarify the aspects of their paper that are clearly different (apart from the “Kitaev” aspect).

Our reply: We have duly cited the work by O. Motrunich, Phys. Rev. B 73, 155115 (2006) in the modified version of our manuscript. This paper discusses the orbital response of a quantum spin liquid with spinon Fermi surface and the possibility of quantum oscillations of this state. While our work focuses on the orbital current in the Kiteav quantum spin liquid and the associated excitations. We feel that the work by O. Motrunich is not directly related to what we discussed in the manuscript.

Reviewer's comment (#2): In the section on “phenomenological picture”, the authors have reproduced a rather standard argument (due originally to Ioffe-Larkin, which they do not cite), which others have used to evaluate the orbital effects (including e.g. the paper mentioned in point 1 above). However, it is unclear why the authors pick a relativistic theory for the b-boson in Eqn. (2). There is no a priori reason for this to be the case–after all not all Mott insulators necessarily have a relativistic gapped boson.
Our reply: We are sorry for the confusion. We did not mean to include the temporal component of the gauge field because it corresponds to screening. The indices μ in Eq. (2) denote the spatial components. We have modified the manuscript to clarify this point. We also cited the paper by Ioffe-Larkin in the revised manuscript in Ref. [47].

Reviewer's wrote: I have not tried to reproduce the detailed results obtained by the authors. They appear to be a natural extension of many previous works, and it seems the authors have done a careful job. While I don’t think the work is necessarily “novel” (by unfortunately a subjective metric), there are possibly readers who can benefit from the results in the manuscript. If the authors address all the concerns by the referees, it could eventually be accepted for publication.
Our reply: We appreciate referee's decision about our work, and believe that the present version addresses all the concerns by the referees.

---

## Editorial Decision

published